# NANSY++: UNIFIED VOICE SYNTHESIS WITH NEURAL ANALYSIS AND SYNTHESIS

**Hyeong-Seok Choi**[1,2], *****Jinhyeok Yang**[2], *****Juheon Lee**[1,2], *****Hyeongju Kim**[2]
[1]Seoul National University
[2]Supertone, Inc.,
{kekepa15,yangyangii,juheon2,hyeongju}@supertone.ai

## ABSTRACT

Various applications of voice synthesis have been developed independently despite the fact that they generate "voice" as output in common. In addition, most of the voice synthesis models still require a large number of audio data paired with annotated labels (e.g., text transcription and music score) for training. To this end, we propose a unified framework of synthesizing and manipulating voice signals from analysis features, dubbed NANSY++. The backbone network of NANSY++ is trained in a self-supervised manner that does not require any annotations paired with audio. After training the backbone network, we efficiently tackle four voice applications - i.e. voice conversion, text-to-speech, singing voice synthesis, and voice designing - by partially modeling the analysis features required for each task. Extensive experiments show that the proposed framework offers competitive advantages such as controllability, data efficiency, and fast training convergence, while providing high quality synthesis. Audio samples: tinyurl.com/8tnsy3uc.

## 1 INTRODUCTION

Most deep learning-based voice synthesis models consist of two parts: generating a mel spectrogram from an acoustic model that takes labeled annotations as input (e.g., text, music score, etc.), and converting the mel spectrogram into a waveform using a vocoder (Wang et al., 2017; Kim et al., 2020; Jeong et al., 2021; Lee et al., 2019; Liu et al., 2022). However, this usually suffers from poor synthesis quality due to the training/inference mismatch of the acoustic model and vocoder. End-to-end training methods has been recently proposed to tackle such issues (Bińkowski et al., 2020; Weiss et al., 2021; Donahue et al., 2021; Kim et al., 2021). Despite the high quality, however, the training process of end-to-end models is often costly, as the waveform synthesis part needs to be trained again when training each different model. Furthermore, regardless of the training strategies of previous studies (end-to-end or not), most of the standard voice synthesis models are not modular enough in that most of the desirable control features are entangled in a single mid-level representation, or so-called latent space. This limits the controllability of such features and restrains the possibility of models being used as co-creation tools between creators and machines. Lastly, although many voice synthesis tasks are analogous in that they are meant to synthesize or control certain attributes of voice, the methodologies developed for each application remain scattered in research fields. These problems call for the need of developing a unified voice synthesis framework.

We stick to three objectives for designing the unified synthesis framework, that is, 1. data-scalable: the training procedure should be done via a minimum amount of labeled dataset while exploiting abundant audio recordings without labels, 2. modular: the training for each application should be done in a modularized way by sharing a universal parameterized synthesizer, 3. high quality: the synthesis quality must persist high standard even by abiding by the modularized training procedure. To this end, we make a core assumption that most of the voice synthesis tasks can be defined by synthesizing and controlling four aspects of voice, that is, pitch, amplitude, linguistic, and timbre. This motivates us to develop a backbone network that can analyze voice into the four properties and then synthesize them back into an waveform. On that account, we propose NANSY++, which

---

*Equal contribution

is improved upon the previous Neural ANalysis and SYnthesis (NANSY) framework (Choi et al., 2021a), by putting forward a new end-to-end self-supervised training method.

First, we propose a self-supervised fundamental frequency ($F_0$) estimation training method that can be trained without any post processing or synthetic datasets. Next, we adopt two self-supervised training methods - information perturbation and bottleneck - to extract a linguistic representation that is disentangled from other representations (Choi et al., 2021a; Qian et al., 2022). Then, we propose to encode timbre information using a content-dependent time-varying speaker embedding, which successfully captures timbre information of unseen target speaker during training. Finally, we propose a high quality synthesis network to convert the 4 analysis representations into a waveform by adopting an inductive bias of human voice production model.

We assume that by exploiting the proposed self-supervised disentangled representation learning strategies on the backbone network, we can encourage several downstream generative tasks to be more data efficient, while not losing the modularity and synthesis quality. Therefore, after training the backbone network, we introduce 4 exemplar applications - voice conversion, text-to-speech (TTS), singing voice synthesis (SVS), voice designing (VOD) - that can be tackled by sharing analysis representations and synthesis network of the backbone. Each application can be simplified and substituted into the task of synthesizing a subset of analysis representations. Through extensive experiments, we verify that NANSY++ enjoys a lot of advantages at once that existing methodologies cannot: high quality output, fast training of modularized application models, data efficiency, and controllability over disentangled voice features.

## 2 NANSY++

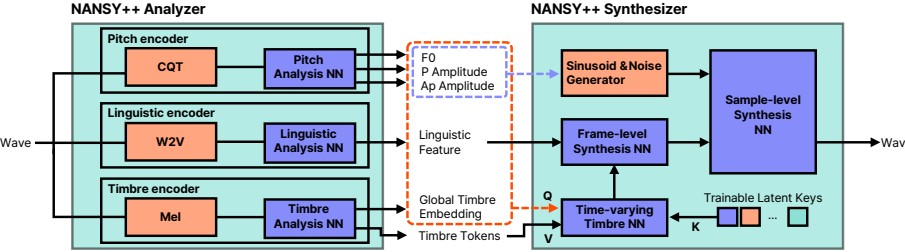

Figure 1: Overview of proposed NANSY++ backbone architecture. All modules in the backbone network is trained in an end-to-end manner within a single analysis and synthesis loop.

### 2.1 SELF-SUPERVISED LEARNING OF PITCH

We represent pitch with fundamental frequency $F_0$ because it is explicitly controllable. In addition, we found that using sinsuoidal signal made by $F_0$ as an input signal for synthesizer can greatly reduce glitches. To train $F_0$ estimator in a self-supervised manner, we first adopt the auto-encoding approach of Engel et al. (2020b). Pitch encoder $f_{\theta_P}$ takes Constant-Q Transform (CQT) as an input feature and outputs probability distribution over 64 frequency bins where it spans from 50Hz to 1000Hz logarithmically (approximately 0.79 semitone per bin). The $F_0$ is estimated by weighted averaging the probability distribution over the frequency bins. The pitch encoder also outputs two amplitude values, periodic amplitude $A_p[n]$ and aperiodic amplitude $A_{ap}[n]$. $F_0[n]$ and $A_p[n]$ are linearly upsampled into a sample-level, $F_0[t]$ and $A_p[t]$, and transformed into a sinusoidal waveform $x[t] = A_p[t] \sin\left(\sum_{k=1}^{t} 2\pi \frac{F_0[k]}{N_s}\right)$, where $N_s$ denotes sampling rate. We also linearly upsample $A_{ap}[n]$ into $A_{ap}[t]$ and generate shaped noise $y[t] = A_{ap}[t] \cdot n[t]$, where $n[t] \sim U[-1, 1]$. Finally the two siganls, $x[t]$ and $y[t]$, are added to form an input excitation signal $z[t] = x[t] + y[t]$ for a synthesizer. After the synthesizer reconstructs the input signal, the pitch encoder is trained with reconstruction loss.

The training solely depending on reconstruction, however, was unstable, and showed poor pitch estimation quality, which was also reported in (Engel et al., 2020b), and as a result they ended up utilizing synthetic audio datasets with pitch labels. While collecting synthetic audio datasets paired with pitch labels is not difficult for musical instruments, it is problematic for speech or singing

voice because we cannot obtain such synthetic datasets. To address this issue, we adopt another self-supervised pitch estimation method that utilizes relative pitch difference between two CQT inputs (Gfeller et al., 2020). First, we extract CQT $\mathbf{X} \in \mathbb{R}^{N \times F}$, where $N$ and $F$ denote the size of time and frequency, respectively. One frequency bin of CQT feature amounts to 0.5 semitone. Next, we crop the frequency axis of $\mathbf{X}$ to obtain two CQT matrices with same size $\tilde{\mathbf{X}}^{(1)}, \tilde{\mathbf{X}}^{(2)} \in \mathbb{R}^{N \times F^{scope}}$. The frequency range of $\tilde{\mathbf{X}}^{(1)}$ consistently spans from $f_{min}$ to $f_{max}$, whereas $\tilde{\mathbf{X}}^{(2)}$ is obtained by randomly shifting the frequency-axis index of $\tilde{\mathbf{X}}^{(1)}$ by $d \sim \mathcal{U}\left(d_{\min}, d_{\max}\right)$. Then, the two cropped CQT features are passed through the pitch encoder and outputs two fundamental frequency sequences $\tilde{F}_0^{(1)}$ and $\tilde{F}_0^{(2)}$. Finally, relative pitch difference loss $\mathcal{L}_{pitch}$ is computed as follows: $\mathcal{L}_{pitch} = h(|\log_2(\tilde{F}_0^{(1)}) - \log_2(\tilde{F}_0^{(2)}) - 0.5d|$, where $h(\cdot)$ denotes huber norm (Huber, 1992). Note that by integrating two self-supervised pitch estimation methods into the end-to-end analysis-synthesis training loop, we achieve absolute pitch estimation without any synthetic datasets. For more detailed CQT configurations and neural architecture of pitch encoder, see Appendix A.1 and A.3.

## 2.2 Disentangled Linguistic Representation

In order to ensure that the linguistic representation is disentangled from other analysis representations, we combine information perturbation method proposed by Choi et al. (2021a) and contrastive loss proposed by Qian et al. (2022). First, a signal is transformed into two different signals using perturbation functions that does not significantly harm the linguistic information. Then, wav2vec features (Babu et al., 2022) extracted from each perturbed signal are passed into linguistic information encoder $f_{\theta_L}$. Finally, the two linguistic representations, $\boldsymbol{L}_n^{(1)}$ and $\boldsymbol{L}_n^{(2)}$, are then used to compute contrastive loss as follows:

$$\mathcal{L}_{\text{contr}} = \sum_{i=1}^{2}\left(-\log\left(\sum_{n=1}^{N} \frac{\exp\left(d\left(\boldsymbol{L}_n^{(1)}, \boldsymbol{L}_n^{(2)}\right)/k\right)}{\sum_{\nu \in \{n\} \cup \mathcal{I}_n} \exp\left(d(\boldsymbol{L}_n^{(i)}, \boldsymbol{L}_\nu^{(i)})/k\right)}\right)\right), \tag{1}$$

where $k$ denotes a temperature parameter, $d$ denotes cosine similarity, and $\mathcal{I}_n$ denotes the set of randomly selected frame indices for negative samples. The intuition of the contrastive loss is that the difference between $\boldsymbol{L}^{(1)}$ and $\boldsymbol{L}^{(2)}$ should be minimized, while the similarity within $\boldsymbol{L}^{(i)}$ is minimized so that the consistent information within an utterance (e.g., timbre information) can be minimized. Note that we pass $\boldsymbol{L}^{(1)}$ to synthesizer so that $f_{\theta_L}$ is jointly trained with reconstruction loss within the whole end-to-end analysis-synthesis loop, which is different from Qian et al. (2022) where they only focused on representation learning for discriminative tasks. As we will show later in section 4.2 and 4.3, the learned linguistic representation is extremely helpful for fast training and data efficiency on synthesis tasks such as TTS and SVS. For more details on information perturbation functions, neural architecture of linguistic information encoder, and hyperparameters for $\mathcal{L}_{\text{contr}}$, see Appendix A.1 and A.3.

## 2.3 Time-Varying Timbre Embeddings

To synthesize waveform from disentangled pitch and linguistic representation, it is important to encode timbre information well enough to fully reconstruct the input signal. Although the timbre information is usually encoded with a single vector, we assume that the capacity of the single vector is not enough to capture the wide variety of timbral characteristics. In addition, we assume that timber is not static but rather varies over time, depending on time-varying contents such as pitch, amplitude, and linguistic information. Therefore, we breakdown timbre features into two - global timbre embedding and timbre tokens - using timbre encoder $f_{\theta_T}$. $f_{\theta_T}$ takes mel spectrogram as an input and produces global timbre embedding $\mathbf{g}$, a single vector summarized with attentive statistical time-pooling (Okabe et al., 2018), which is expected to capture overall timbral characteristics within an utterance. The timbre tokens $\boldsymbol{v}_i$, $i = 1, ..., I$, are expected to capture diverse timbral characteristics into a fixed number of tokens. The timbre tokens are extracted by adopting a cross-attention mechanism introduced by Jaegle et al. (2021), where trainable latent vectors are used as queries, and key and value are extracted from timbre encoder as illustrated in Appendix A.3, Fig. 8.

After extracting timbre features, we adopt another cross-attention mechanism by Yin et al. (2022) to extract time-varying timbre embeddings that depends on other analysis content representations.

Specifically, we used the concatenation of $[F_0, A_p, A_{ap}, \boldsymbol{L}, \mathbf{g}]$ as a query. Keys are composed of $I$ trainable vectors and the timbre tokens are used as values. Finally, the output of the cross-attention mechanism and $\mathbf{g}$ is interpolated using spherical linear interpolation to make time-varying timbre embeddings. For more details of time-varying timbre embeddings, see Appendix A.3.

## 2.4 Waveform Synthesis

The synthesis modules are composed of two synthesizers, frame-level and sample-level synthesizer. The frame-level synthesizer first takes linguistic feature and time-varying speaker embeddings to produce a frame-level condition for a sample-level synthesizer. After that, the frame-level condition are linearly upsampled into sample-level features for sample-level conditioning. The sample-level synthesizer then takes the excitation signal and sample-level features to reconstruct an input waveform. For the sample-level synthesizer architecture, we adopt the generator architecture of Parallel WaveGAN (Yamamoto et al., 2020). The whole analysis and synthesis modules are then trained with reconstruction loss. For the reconstruction loss, we used multi-scale spectrogram (MSS) loss (Wang et al., 2019) and mel spectrogram loss (Kong et al., 2020). Note that it was crucial to use linear frequency scale spectrogram for MSS loss rather than log-scale spectrogram for the stable training of pitch encoder, which shares similar observations to what Turian & Henry (2020) has reported. We also used adversarial loss and feature matching loss for high quality synthesis. Multi-period discriminator (MPD) was used as a discriminator architecture (Kong et al., 2020). For more details of synthesizers, please refer to Appendix. Note that we downsample input waveforms into 16 kHz before passing into analysis modules and let the synthesizer produces the original 44.1 kHz waveforms. This enables 16 kHz to 44.1 kHz audio upsampling.

## 3 Experiments on Backbone

We trained the backbone model on 10,571 hours (speech: 10,092 hours, singing: 479 hours) of proprietary 44.1 kHz audio recordings composed of 6,176 speakers and 624 singers. We trained the backbone model for 1M iterations with Adam optimizer (Kingma & Ba, 2014) with the learning rate of $10^{-4}$. The learning rate for MPD was $2 \times 10^{-4}$. The batch size was set to 60 using 10 RTX 3090 GPUs.

## 3.1 Fundamental Frequency Detection

To evaluate the robustness of the proposed pitch encoder in harsh conditions, we tested the performance of the pitch encoder on noisy speech testset. We sampled 30 speech and noise recordings from the VCTK and DEMAND dataset (Veaux et al., 2017; Thiemann et al., 2013), respectively, and mixed them with 5 dB signal-to-noise ratio (SNR). For the baseline

Table 1: ABX results (%)

| praat | rapt | pyin | crepe |
|-------|------|------|-------|
| 89.8 | 88.8 | 84.6 | 71.1 |

models, we chose 4 popular $F_0$ estimators, that is, praat, rapt, pyin, and crepe (Boersma et al., 1993; Boersma & Van Heuven, 2001; Jadoul et al., 2018; Talkin & Kleijn, 1995; Mauch & Dixon, 2014; Kim et al., 2018). We conducted ABX test, where 15 participants were asked which of the two samples (A and B) sounds more similar to the original sample (X). One of the two samples was generated by reconstructing the original sample using the NANSY++ backbone, and another sample was reconstructed by simply replacing only the original $F_0$ sequence into the ones that were extracted from one of the 4 baseline models. We conducted the experiments on Mechanical Turk (MTurk). The results are shown in Table 1. The results clearly show that $F_0$ estimated by NANSY++ pitch encoder outperforms other pitch estimators significantly.

## 3.2 Reconstruction

In order to use the backbone synthesizer as a synthesis module for various applications, it is important to test the reconstruction performance. We randomly selected 5 audio recordings for each of 25 unseen speakers and 25 unseen singers during training. 20 participants were involved for the evaluation on MTurk. As a baseline model ($\mathbf{BL}_H$), we chose HiFi-GAN as it is the most widely used mel-to-waveform converter (Kong et al., 2020). We trained HiFi-GAN using the same training set to NANSY++. For

Table 2: Reconstruction results.

| | Speech | Singing |
|---|--------|---------|
| GT | $4.39_{\pm 0.05}$ | $4.35_{\pm 0.05}$ |
| $\mathbf{BL}_H$ | $4.00_{\pm 0.06}$ | $3.26_{\pm 0.06}$ |
| **Ours** | $\mathbf{4.37}_{\pm 0.05}$ | $\mathbf{4.36}_{\pm 0.05}$ |

a fair comparison, we doubled the upsampling rate of the penultimate block of the HiFi-GAN generator, so that it can produce 44.1kHz waveform. The results are shown in Table 2. The results clearly show that NANSY++ synthesizer produces better waveforms for both speech and singing voice. Especially, significant improvement was observed on singing voice.

## 4 APPLICATIONS

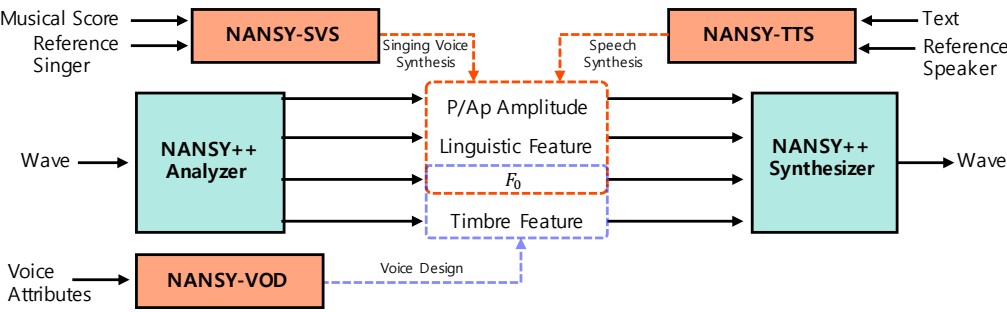

Figure 2: Overview of exemplar applications integrated into the NANSY++ backbone architecture. Each application can be substituted into a problem of estimating analysis features from the backbone.

We introduce 4 exemplar applications that can be integrated with NANSY++. Various conditional generative models can be integrated by generating analysis features from conditions.

### 4.1 VOICE CONVERSION

We perform zero-shot voice conversion by simply replacing the original timbre features into the timbre features extracted from a single utterance of a target speaker. To further match the style of the target speaker, we also transformed the $F_0$ statistics of the source speaker into the target speaker's. As a baseline model ($\mathbf{BL_Y}$), we selected the state-of-the-art zero-shot voice conversion model, YourTTS (Casanova et al., 2022). Note that YourTTS requires paired (text, audio) dataset to train the model. Other than the NANSY++ trained with the entire

Table 3: Voice conversion results.

| MODEL | MOS | SSIM |
|---|---|---|
| TGT as TGT | $4.08_{\pm 0.05}$ | $3.47_{\pm 0.06}$ |
| $\mathbf{BL_Y}$ | $3.23_{\pm 0.06}$ | $2.78_{\pm 0.12}$ |
| **Ours** (*vctk*) | $3.72_{\pm 0.06}$ | $2.99_{\pm 0.12}$ |
| **Ours** (*entire*) | $\mathbf{3.79_{\pm 0.05}}$ | $\mathbf{3.16_{\pm 0.10}}$ |

dataset (**Ours** (*entire*)) explained in section 3, we trained additional model solely on VCTK dataset (**Ours** (*vctk*)) to match the training dataset setting of the baseline model. We tested each model using 11 unseen speakers during training following EXP. 1 of Casanova et al. (2022). Each speaker was converted into 4 different speakers, resulting in 44 conversion pairs. For each source-target conversion pair, we randomly selected 5 utterances, resulting in 220 audio samples in total. We tested the naturalness (MOS) and speaker similarity (SSIM) following Wester et al. (2016). The evaluation was done by 20 participants on MTurk. Note that we downsampled the 44.1 kHz NANSY++ outputs into 16 kHz for a fair comparison to YourTTS as it only supports 16 kHz synthesis. The results are shown in Table 3. TGT as TGT denotes the setting where the target speakers are compared with their own utterances. The results clearly show that NANSY++ can change the voice with better naturalness and speaker similarity then the baseline model even though the model was trained without any text transcripts paired with audio recordings. In addition, we could achieve a noticeable speaker similarity improvement when trained with more audio recordings, showing the data scalability of the training method of NANSY++.

### 4.2 TEXT-TO-SPEECH

In this section, we describe NANSY-TTS, a text to speech (TTS) application utilizing NANSY++ framework. NANSY-TTS is independently trained using NANSY++ analysis features. We show that this modular training approach is data efficient, while maintaining high-fidelity synthesis quality.

### 4.2.1 METHOD

**Architecture**  NANSY-TTS consists of a phoneme encoder, style encoder, linguistic decoder, $F_0$ decoder, amplitude decoder, and length regulator. All modules operate in a non-autoregressive manner. By utilizing the timbre encoder of NANSY++ analyzer, there is no need to consider the timbre representation for NANSY-TTS. Therefore, the style encoder only handles the prosody (e.g., speaking pace, pitch, and loudness). The output style vector is used for duration predictor and all decoders. More details and the overview figure can be found at Appendix B.1.

**Training and inference**  To train the proposed TTS module, we optimize Mean Absolute Error(MAE) for linguistic features, $F_0$, $A_p$, and $A_{ap}$ extracted from NANSY++ analyzer. The aligner is trained independently to other modules in NANSY-TTS. The duration predictor is optimized by minimizing MAE for the duration of each phoneme using the alignment calculated by the monotonic alignment algorithm (MAS) (Kim et al., 2020) with the aligner module. In the inference stage, the TTS module first generates linguistic feature, P/Ap amplitudes, and $F_0$. Then, NANSY++ synthesizer takes these generated features and timbre features to generate an waveform. More details can be found at Appendix B.2

### 4.2.2 EXPERIMENTS

**Data efficiency**  To evaluate data efficiency, we used a single speaker (reader id: 8051) from the Hi-Fi multi-speaker english TTS dataset (Bakhturina et al., 2021). The dataset has 30 hours of speech audio sampled at 44.1kHz. We made three subsets with data of 5, 10 and 30 minutes, and the larger subsets include small subsets. First, we measured character error rate (CER) to evaluate the intelligibility of the speech according to

Table 4: TTS data efficiency evaluation.

| MODEL | MOS | CER (%) |
|---|---|---|
| GT | $4.47_{\pm 0.09}$ | 1.93 |
| $\mathbf{BL}_G$ / **Ours** (5) | $1.37_{\pm 0.07}$ / $3.31_{\pm 0.10}$ | 42.50 / 3.29 |
| $\mathbf{BL}_G$ / **Ours** (10) | $2.07_{\pm 0.11}$ / $3.39_{\pm 0.10}$ | 18.59 / 3.05 |
| $\mathbf{BL}_G$ / **Ours** (30) | $2.78_{\pm 0.11}$ / $3.64_{\pm 0.10}$ | 5.34 / 2.20 |
| $\mathbf{BL}_G$ / **Ours** ($full$) | $3.38_{\pm 0.11}$ / $\mathbf{4.07}_{\pm 0.09}$ | 2.65 / **1.68** |

the size of the dataset. The synthesized speech from the 100 test sentences were transcribed by the pretrained speech recognition model (Silero, 2021). As a baseline model ($\mathbf{BL}_G$), the official implementation of Glow-TTS was used as an acoustic model. We used the same 44.1 kHz HiFi-GAN as a baseline vocoder mentioned in section 3.2. The training configuration of NANSY-TTS (**Ours**) was set to be the same as that of the baseline model. As shown in Table 4, NANSY-TTS preserved relatively high MOS and low CER regardless of the size of the dataset, whereas evaluation results of the baseline model showed significantly dropped performance as the size of the dataset became smaller. In particular, NANSY-TTS trained with the 30 minutes subset outperforms the baseline model trained with the full dataset. This indicates the proposed modular training approach offers a significant data efficiency for TTS with high synthesis quality.

**Zero-shot TTS**  The setting for this evaluation is the same as section 4.1, but with three differences. First, the model generates speech samples using one randomly selected reference speech per speaker as the style input. Second, we randomly selected 9 utterances for each of 11 unseen speakers, resulting in 99 audio samples in total. Finally, we trained NANSY-TTS on VCTK using two versions of NANSY++ as mentiond in section 4.1, respectively. As shown in Table 5, NANSY-

Table 5: Zero-shot TTS results.

| MODEL | MOS | SSIM |
|---|---|---|
| TGT as TGT | $3.98_{\pm 0.07}$ | $2.91_{\pm 0.09}$ |
| $\mathbf{BL}_Y$ | $3.33_{\pm 0.08}$ | $2.21_{\pm 0.09}$ |
| **Ours** ($vctk$) | $4.27_{\pm 0.07}$ | $2.55_{\pm 0.09}$ |
| **Ours** ($entire$) | $\mathbf{4.28}_{\pm 0.07}$ | $\mathbf{2.65}_{\pm 0.09}$ |

TTS outperforms the baseline. Interestingly, NANSY-TTS even achieved significantly better MOS than actual recordings. We conjecture that this is because some speakers in the VCTK datasets speak in rather unnatural prosody, while NANSY-TTS learns how to speak in natural prosody given the text transcription. We discuss the details in the Appendix B.3 further.

### 4.3 SINGING VOICE SYNTHESIS

In this section, we describe a singing voice synthesis (SVS) application that generates NANSY++ analysis features from a musical score, namely NANSY-SVS. The two major challenges of SVS are (i) limited amount of labeled data and (ii) unpleasant glitches of synthesized voice. The first problem stems from the difficulty of dataset collection process. Labeling singing data often demands

more human labor and time compared to speech data. Another issue is that generating high-fidelity singing voice from mel spectrogram is well known to be difficult owing to a wide pitch range, long continuous pronunciation, and high sampling rate (Chen et al., 2020a; Perrotin et al., 2021; Morrison et al., 2022). To tackle these problems, we introduce NANSY-SVS that builds upon the benefits of NANSY++ framework. NANSY-SVS addresses both concerns regarding the dataset size and high-fidelity synthesis by modeling the disentangled NANSY++ analysis features.

### 4.3.1 METHOD

**Architecture** NANSY-SVS uses a phoneme sequence, MIDI-pitch sequence, and global timbre embedding (from NANSY++ analyzer) as conditional inputs to generate linguistic features and $F_0$ contours in an autoregressive manner. The linguistic feature at the $t$-th frame $\boldsymbol{L}_t$ is inferred from $\boldsymbol{L}_{<t}$, phoneme sequence, MIDI-pitch sequence, and singer-ID embedding. For the $F_0$ contour, we predicted the difference (residual-$F_0$) between input MIDI-pitch and target $F_0$ contour instead of modeling it directly. The residual-$F_0$ range ending in -1200 cents to 1200 cents is quantized in 100 cents increments, yielding 241 possible values. The 241-dimensional residual-$F_0$, $RF_t$, was inferred from $RF_{<t}$, MIDI-pitch sequence, phoneme sequence, and singer-ID embedding. The amplitude predictor module infers P/Ap amplitudes from the generated linguistic features and $F_0$ contour. More details of NANSY-SVS can be found in Appendix C.1.

**Training and inference** To train NANSY-SVS, we optimize MAE for linguistic features, P/Ap amplitude, and cross-entropy for residual-$F_0$ classification. At inference time, linguistic features and residual-$F_0$ are first generated in autoregressive way, and then P/Ap amplitudes are inferred from the generated results. Finally, the generated features are passed through the NANSY++ synthesizer along with the timbre features extracted from the reference audio to produce a final waveform.

### 4.3.2 EXPERIMENTS

**Experimental setup** We trained the model using OpenCPOP dataset (Wang et al., 2022), a public singing dataset of 100 songs from a single female singer with aligned phoneme and MIDI-pitch sequence. We followed the official train/test split of Wang et al. (2022), i.e. 95 songs and 5 songs for training and evaluation, respectively. To verify that our proposed model can be trained to synthesize high-quality singing voice using a limited amount of training data, we constructed a subset by extracting 80, 160, and 320 segments from the original dataset consisting of 3550 segments in total. This corresponds to 10%, 5%, and 2.5% of the entire dataset, and we trained 4 models in total using each subset. Note that each subset was constructed to include all phonemes and MIDI-pitch at least once. As a baseline model ($\mathbf{BL}_{\mathrm{D}}$), we chose Diffsinger (Liu et al., 2022), which showed state-of-the-art performance with a diffusion-based model structure. We trained the baseline model with the same setting as NANSY-SVS using the official implementation for both acoustic model and vocoder.

**Evaluation** We conducted a listening test to evaluate the quality of generated singing voice. We randomly selected 12 segments from each of the 5 test songs, and generated 60 singing voice segments in total for each model. The singing-ID embedding was obtained from a randomly sampled 30-second segment in the training set. 15 participants were asked to assess the naturalness of each audio on a 5-point scale. Table 6 shows that the perceptual quality of the NANSY-SVS outperforms the base-

Table 6: SVS data efficiency evaluation.

| MODEL | MOS |
|---|---|
| $\mathbf{BL}_{\mathrm{D}}$ / **Ours** $_{(80)}$ | $2.67_{\pm 0.08}$ / $3.81_{\pm 0.06}$ |
| $\mathbf{BL}_{\mathrm{D}}$ / **Ours** $_{(160)}$ | $3.05_{\pm 0.08}$ / $3.86_{\pm 0.05}$ |
| $\mathbf{BL}_{\mathrm{D}}$ / **Ours** $_{(320)}$ | $3.49_{\pm 0.06}$ / $3.85_{\pm 0.05}$ |
| $\mathbf{BL}_{\mathrm{D}}$ / **Ours** $_{(full)}$ | $3.83_{\pm 0.06}$ / $\mathbf{3.87_{\pm 0.06}}$ |
| GT / Recon. | $4.026_{\pm 0.056}$ / $4.031_{\pm 0.053}$ |

line in all training conditions. As the size of the training data decreased, the MOS of Diffsinger decreased by up to 1.16 while the MOS of NANSY-SVS decreased by only 0.06. From this, we confirm that the proposed modularized training method with NANSY++ backbone is efficient in data-limited training condition. Other experiments on SVS can be found at Appendix C.2.

## 4.4 VOICE DESIGNING

To further explore the usability of disentangled NANSY++ features, we tackle the challenging problems of manipulating voice such as creating new voice identities and editing voice attributes. Previous works have investigated extrapolating of speaker embedding space using Gaussian mixture models (GMMs) (Bilinski et al., 2022) or learning a speaker embedding prior within a variational

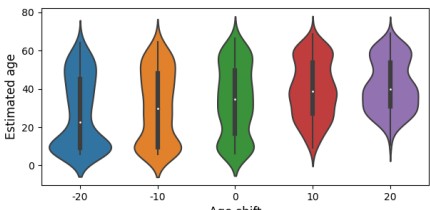

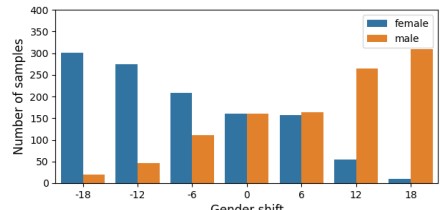

Figure 3: Age shift evaluation.        Figure 4: Gender shift evaluation.

autoencoder (VAE) framework (Stanton et al., 2022) to synthesize new speakers. In this section, we develop a voice designing system, namely NANSY-VOD, that can both change voice attributes and generate novel voice identities. NANSY-VOD also enables fine-grained control over voice manipulation in an interpretable way by leveraging speaker's gender and age as conditioning variables.

### 4.4.1 METHOD

**Model** NANSY-VOD consists of three normalizing flow networks that learn the distribution of $F_0$ statistics (e.g., median and variance), a global timbre embedding, and timbre tokens, respectively. The first network estimates the conditional distribution of the $F_0$ statistics given gender and age, and is trained according to the maximum likelihood. The second network is optimized to maximize the likelihood of the global timbre embedding conditioned on the $F_0$ statistics, gender, and age. Similarly, the third network models the distribution of the timbre tokens given the global timbre embedding, the $F_0$ statistics, gender, and age. Since the timbre tokens consist of multiple column vectors lying in a common feature space, we choose to use each vector as individual input to the network. To distinguish these vectors, a token index is used as an additional conditioning term. By sharing parameters, the third network is trained more efficiently compared to the naive implementation that models the entire timbre tokens at once. More details can be found in Appendix D.1.

**Continuous gender** We use continuous real values instead of binary values to represent gender. The intuition is that (i) voices of pre-pubertal children may not be gender distinct, and (ii) some people's voices sound neutral. By allowing relaxed labels for gender, we expect that the gradual control of the gender attribute of voice can be implemented at inference time. To get continuous gender labels, another network called SPNet is employed and more details can be found at Appendix D.4.

**Training and inference** Input data is obtained from NANSY++ analysis modules and conditioning data (i.e. gender and age) is labeled by SPNet for training. The negative log-likelihoods of the three networks are summed to compute the total loss and NANSY-VOD is optimized in the same way as a typical normalizing flow. At inference time, NANSY-VOD supports two functionalities: voice attribute control and new voice generation. To edit voice attributes, first, the inputs are transformed to flow latent variables conditioned on the source speaker's gender and age. The edited NANSY++ features are then retrieved by converting the latent variables back conditioned on different age and gender. To create a new voice identity, latent variables are sampled from the normal distribution and transformed via the inverse pass of NANSY-VOD with target age and gender.

### 4.4.2 EXPERIMENTS

**Experimental setup** To evaluate the proposed method, we used the NANSY++ backbone dataset and three AI-Hub conversation datasets (https://aihub.or.kr): kids, adults, and elderly people. The AI-Hub datasets consist of approximately 6821 hours of Korean recordings with gender and age labels. There are a total of 6001 speakers whose ages range from 3 to 91. We randomly selected 4800 and 600 speakers, and constructed training and validation sets respectively by merging their utterances and speech data of the NANSY++ backbone dataset. From the remaining 601 speakers, we prepared a set of 320 unseen speakers by sampling 40 male and female speakers from each age group of (0, 10], (10, 30], (30, 50], and (50, 70]. We trained NANSY-VOD for about 250K iterations with the AdamW optimizer (Loshchilov & Hutter, 2017) of learning rate $2 \times 10^{-4}$. A batch size was set to 256 for each of four RTX 3090 GPUs. Other experimental settings such as model architecture and hyperparameters are attached at Appendix D.1.

**Voice attribute control** NANSY-VOD controls voice attributes by using different conditioning inputs for the forward and inverse processes of normalizing flows. First, we edited the age attributes

of the 320 unseen speakers by adding or subtracting different values to their original ages. The synthesizer reconstructed audio using the linguistic feature of a source speaker and the other edited features. Then, the ages of the converted voices were estimated by SPNet. We report the estimated age distributions of the original voices and the edited voices in Fig. 3. A shift in the age distribution was observed as the target age shifts, supporting the age controllability of NANSY-VOD.

We also edited the gender attribute and investigated the gender of the converted voices using SPNet. Since NANSY-VOD uses a continuous gender, SPNet was used to label the gender of a source speaker. Fig. 4 shows the gender distributions of the original and converted voices. We observed the gender distribution varies as we shift the gender values accordingly. This shows that NANSY-VOD can transform a voice into a more masculine or feminine voice by controlling the gender attribute.

**Voice identity generation** We evaluated the performance of voice identity generation in terms of speaker diversity. To measure how diverse speakers NANSY-VOD can generate, we utilized the speaker distance metric proposed by Stanton et al. (2022), which is reformulated as $\underset{i}{\mathrm{median}} \min_{j \neq i} d(V_i, V_j)$, where $d(\cdot, \cdot)$ denotes the cosine distance and $V_i$ represents the speaker

Table 7: Speaker diversity. $n_s$ denotes the number of speakers in comparison set.

| $n_s$=320 | $n_s$=240 | $n_s$=160 | $n_s$=1 | Ours |
|-----------|-----------|-----------|---------|------|
| 0.51 | 0.46 | 0.39 | 0.17 | 0.46 |

feature of the $i$-th utterance extracted from ECAPA-TDNN (Desplanques et al., 2020). We synthesized 320 speakers from NANSY-VOD and synthesized audio using the linguistic features of the utterances in the test set. For comparison, we constructed four sets of 320 utterances consisting of different numbers of speakers (320, 240, 160, and 1). Table 7 shows the speaker diversities of the set generated by NANSY-VOD and the other sets. The set consisting of 320 speakers records the highest diversity and we use this score as an estimate of the speaker diversity of the real speakers. As the number of speakers in the set decreases, the speaker diversity gets reduced. The speaker diversity of NANSY-VOD is close to that of the 240 speakers set. This implies that NANSY-VOD can generate various speakers having 75% diversity compared to the real speaker distribution.

## 5 RELATED WORKS

Self-supervised representation training methods for speech has recently been widely studied in the goal of training rich representations mostly for downstream tasks such as automatic speech recognition (Baevski et al., 2020; Babu et al., 2022; Hsu et al., 2021; Huang et al., 2022). There has also been studies that utilized the self-supervised speech representation on synthesis task such as voice conversion (Lin et al., 2021; hao Lin et al., 2021). For TTS applications, Siuzdak et al. (2022) proposed to employ wav2vec 2.0 feature as a mid-level representation of a TTS model. However, they used the representation from wav2vec 2.0 network finetuned on English text transcriptions, hence might not be sufficient for training on languages without labeled datasets. Kim et al. (2022) has proposed to apply transfer learning framework using pseudo phoneme sequence extracted from wav2vec 2.0 feature and showed that the model can be trained using a small amount of dataset.

Previous works have shown that state-of-the-art vocoder such as HiFi-GAN produces glitches, especially on long notes (Morrison et al., 2022; Chen et al., 2020a). Morrison et al. (2022) showed that an autoregressive neural architecture can reduce this issue. Another line of vocoders are the ones that take sinusoidal signal as a network input (Wang et al., 2019; Hono et al., 2021). These types of vocoders do not exhibit such glitches on long notes even without the autoregressive architecture.

## 6 CONCLUSION

In this work, we proposed a unified voice synthesis framework NANSY++ that can analyze signal into disentangled representations and synthesize high-quality waveform. Because the training only requires audio recordings, we can achieve data scalability. We have also proposed to integrate various voice synthesis applications in a modularized way. By integrating the applications into the centralized backbone network, we found that we can train models for each application using only a small amount of dataset. Although each module is not trained in an end-to-end manner, the proposed modularized training strategy does not suffer from the common performance drop phenomenon induced by training/inference mismatch. We believe that the proposed framework provides useful representations for synthesis tasks. Therefore, we hope that the proposed framework will help to move away from the current predominant training strategy (*text-to-mel* acoustic model & *mel-to-wav* vocoder) and expand the scope of research.

## 7 Reproducibility Statement

Details of neural architectures for every models used in this paper are described in Appendix. We have also included every hyperparameters used for training in Appendix.

## 8 Ethics Statement

There is a possibility of the proposed method being used with harmful intentions by anonymous users. Because the proposed method can clone arbitrary voice in a zero-shot manner, we believe the technology should be released only to the identified and authorized users.

As the voice synthesis technologies advance rapidly, attempts have been made to counteract the concerns surrounding the harmful intents of the voice synthesis technology. Recently, anti-spoofing challenges have drawn attentions from many fields because of its importance, and showed promising results on anti-spoofing for speech (Yamagishi et al., 2021). We hope to see more domain experts to engage and develop a voice verification system to prevent abuse of fake voice.

Despite the concerns on negative consequences of the technology, there is also a bright side of such technologies when used properly. One such example is when used as an interactive tool between humans and machines. Because the proposed methods enable controls on voice, we believe it can foster human creativity in various aspects.

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

# A BACKBONE

## A.1 TRAINING DETAILS

**Linguistic feature** In the training stage of the backbone architecture, it is important to perturb the information before extracting the linguistic feature so that it does not contain any information related to timbre. We adopted the four perturbation functions - i.e. formant shift, pitch randomization, parametric equalizer - used in Choi et al. (2021a) along with two more perturbtion functions, breathiness perturabtion, and additive noise. The noise was randomly selected from the DEMAND dataset (Thiemann et al., 2013).

We decide indice set $\mathcal{I}_n$ for negative samples by generating random mask that does not contain the nearby 10 frames of the linguistic feature so that the negative frames are not sampled from the ones that contain the same linguistic characteristics to the positive frame. We set the temperature $k$ to 0.1 and the coefficient for $L_{contr}$ was linearly scheduled from $10^{-5}$ to 10, following (Qian et al., 2022).

**Self-supervised $F_0$ estimation** The configurations for CQT $\mathbf{X} \in \mathbb{R}^{N \times F}$ and cropped CQT $\tilde{\mathbf{X}}^{(1)}$, $\tilde{\mathbf{X}}^{(2)} \in \mathbb{R}^{N \times F^{scope}}$ are as follows. The number of bins per octave is set to 24. The frequency range spans from 32.7 Hz to 8000 Hz. The size of frequency-axis of CQT feature before crop $F$ is 191. The frequency size of the cropped CQT features $F^{scope}$ is 160. The minimum and maximum bound for sampling distribution $d \sim \mathcal{U}(d_{\min}, d_{\max})$ for index shift was set to -12 and +12, which amounts to -6 and +6 semitone.

## A.2 QUALITATIVE RESULTS ON PITCH ESTIMATION

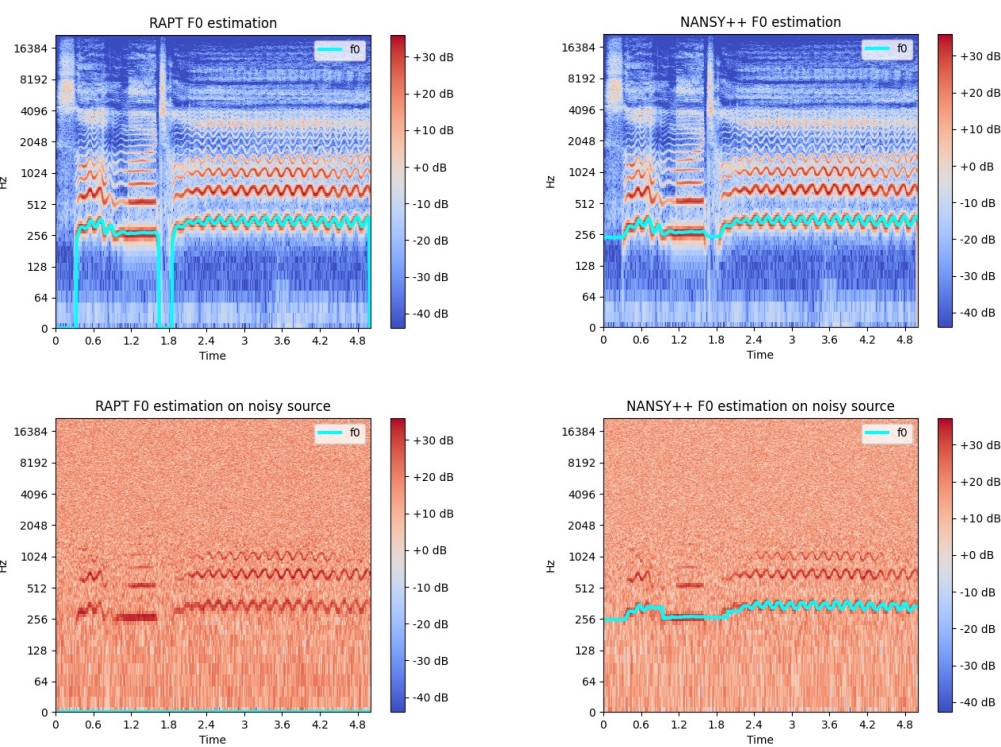

Figure 5: The qualitative results on $F_0$ estimation. Left column shows the $F_0$ estimation results from rapt algorithm. Right column shows the $F_0$ estimation results from NANSY++ pitch encoder. First row shows the $F_0$ estimation results from clean signal. Second row shows the $F_0$ estimation results from noisy signal. The results clearly demonstrates that the pitch encoder is indeed estimating $F_0$, even robustly for noisy signal.

In order to show that the output from the pitch encoder is really estimating fundamental frequency, we show the results of the estimated fundamental frequency in Figure 5. We can see that for clean signals, rapt algorithm and NANSY++ pitch encoder are estimating the same $F_0$ trajectory, showing that the pitch encoder trained in a self-supervised manner is actually estimating $F_0$. Additionally, we have also tested the algorithms on a noisy signal to demonstrate the robustness of the proposed pitch encoder on noise. The second row shows that NANSY++ pitch encoder can estimate $F_0$ even in the harsh environment, whereas rapt algorithm completely fails.

### A.3   NEURAL ARCHITECTURES

**Pitch encoder**   For the architectural design of pitch encoder, we used 1D-CNN blocks operating in frequency-axis combined with bi-directional gated recurrent unit similar to Choi et al. (2021b). The architecture is illustrated in Fig. 6. For the last activation function of P/Ap amplitude heads, we used exponential sigmoid used in Engel et al. (2020a).

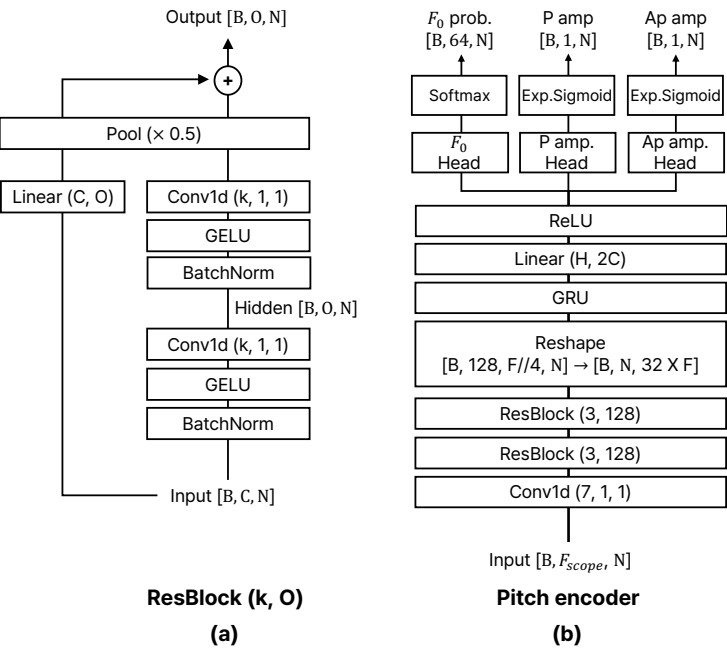

Figure 6: Pitch encoder architecture. k, d, s in Conv1d (k, d, s) denotes kernel size, dilation, and stride. k, O in ResBlock (k, O) denotes kernel size and output channel size. C, O in Linear (C, O) denote input and output channel size.

**Linguistic information encoder**   For the architectural design of linguistic information encoder, we used Convolutional Gated Linear Unit (ConvGLU) blocks (Dauphin et al., 2017). The architecture is illustrated in Fig. 7.

**Time-varying Speaker Embeddings**   The architectural details of timbre encoder and time-varying timbre encoder are shown in Fig. 8. We borrowed the neural architecture for timbre encoder from ECAPA-TDNN (Desplanques et al., 2020). The timbre tokens are extracted via cross-attention mechanism, where 50 trainable latent vectors are used a query, and features from multilayer feature aggregation (MFA) block of ECAPA-TDNN is transformed into key and value. We also extract global timbre embedding via attentive statistical pooling (ASP). Time-varying timbre encoder then utilizes another cross-attention mechanism by taking content queries, a concatenation of $[F_0, A_p, A_{ap}, \boldsymbol{L}, \mathbf{g}]$, trainable 50 latent keys, and timbre tokens as values. The output of the cross-attention and global timbre embedding is then interpolated on a spherical surface to produce the final time-varying timbre embeddings.

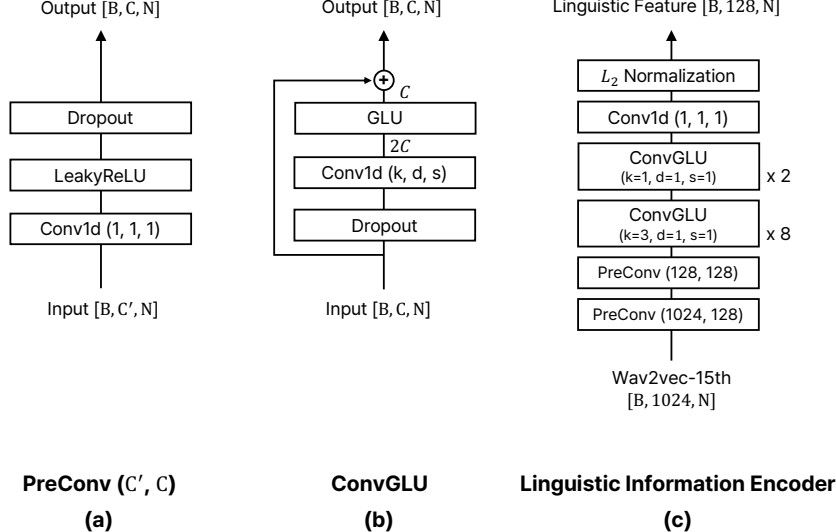

Figure 7: Linguistic information encoder architecture. C' and C in PreConv denote input channel and output channel size. k, d, s in Conv1d (k, d, s) denotes kernel size, dilation, and stride.

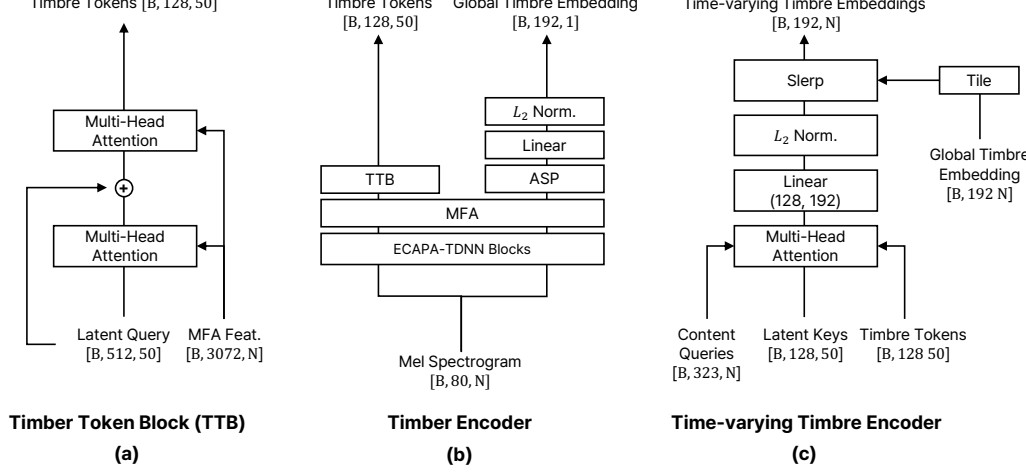

Figure 8: The architecture of timbre encoder and time-varying timbre encoder. MFA denotes Multilayer Feature Aggregation block of ECAPA-TDNN. ASP denotes Attentive Statistical Pooling.

**Synthesizer** The architectural details of frame-level and sample-level synthesizers are shown in Fig. A.3. The frame-level synthesizer takes the linguistic feature and time-varying speaker embeddings and outputs frame-level condition. The frame-level condition is then passed through the sample-level synthesizer along with $F_0$, periodic amplitude (P amp), aperiodic amplitude (Ap amp) to produce a waveform. We borrowed the neural architecture of parallel wavegan (PWGAN) for the sample-level synthesizer (Yamamoto et al., 2020).

## B TEXT-TO-SPEECH

### B.1 DETAILED ARCHITECTURE

**Encoder** The architectural details of encoders are shown in Fig. 11. Similar to Min et al. (2021), NANSY-TTS uses the style encoder and conditional layer normalization (cLN) to train the various prosody. However, we use the hidden feature extracted from the third layer of wav2vec network

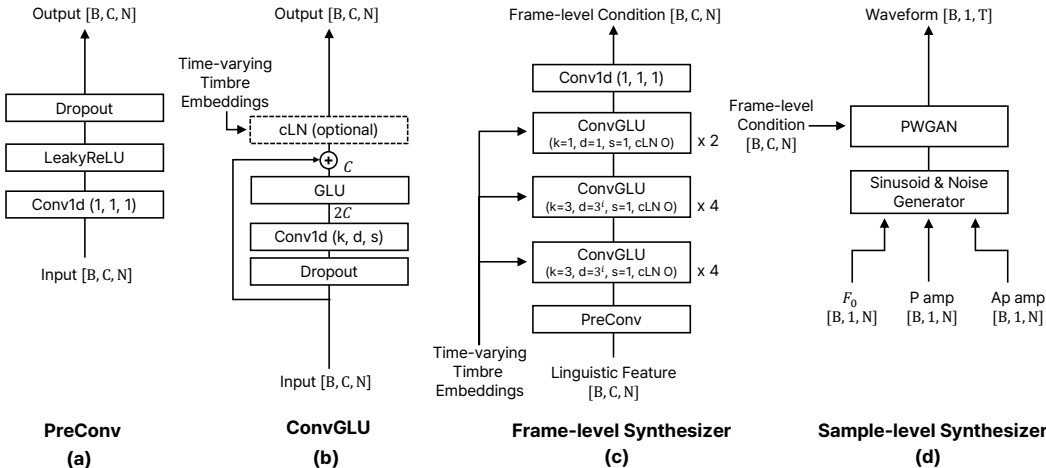

Figure 9: The architecture of frame-level synthesizer and sample-level synthesizer. cLN denotes conditional Layer Normalization.

(Babu et al., 2022) instead of mel spectrogram as the style input. The output vector encoded by the style encoder is used by all cLNs in NANSY-TTS. The phoneme encoder is composed of a lookup embedding table, 3 ConvReLUNorm blocks, 3 Transformer(Vaswani et al., 2017) blocks, and a final linear layer with LeakyReLU.

**Decoder** The architectural details of decoders are shown in Fig. 12. First, the phoneme feature sequences are upsampled using the duration predicted by the duration predictor or extracted by the aligner. The upsampled sequences are then fed to all decoders. The amplitude decoder is composed of 3 ConvReLUNorm blocks with cLNs and a final linear layer. To predict $F_0$ more elaborately, the $F_0$ decoder also use the hidden features from the amplitude decoder as an additional input. The $F_0$ decoder consists of 2 ConvReLUNorm blocks with cLNs, GRU and a final linear layer.

**Aligner** We train an aligner independently of the TTS model. It consists of an encoder and a decoder. The encoder is composed of 5 ConvReLUNorm(k=3,3,3,1 and 1) blocks as shown in Fig. 11. The decoder is the same as in Kim et al. (2020) except for three differences. First, it has only 2 blocks instead of 12 blocks of the original version. Secondly, it uses the linguistic feature as the output instead of mel spectrogram. Finally, the aligner including the decoder is only used during training, and the value calculated by MAS is used as the phoneme duration.

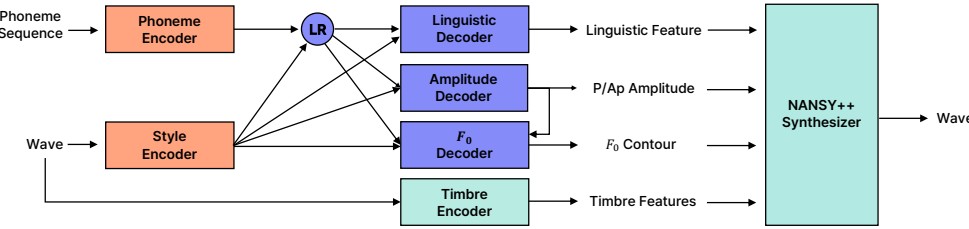

Figure 10: Overview of NANSY-TTS.

## B.2 TRAINING DETAILS

All models were trained for 200K iterations using Adam optimizer (Kingma & Ba, 2014) with the learning rate of $10^{-4}$. In the training stage, we use randomly sliced ground truth waveform as the style reference input. This helps to ensure robust performance even when a nonparallel reference, which has a different content from the input text, is used as the style input in the inference stage. To train the prosody components efficiently, we apply the min-max normalization for $A_p$, $A_{ap}$ and $F_0$.

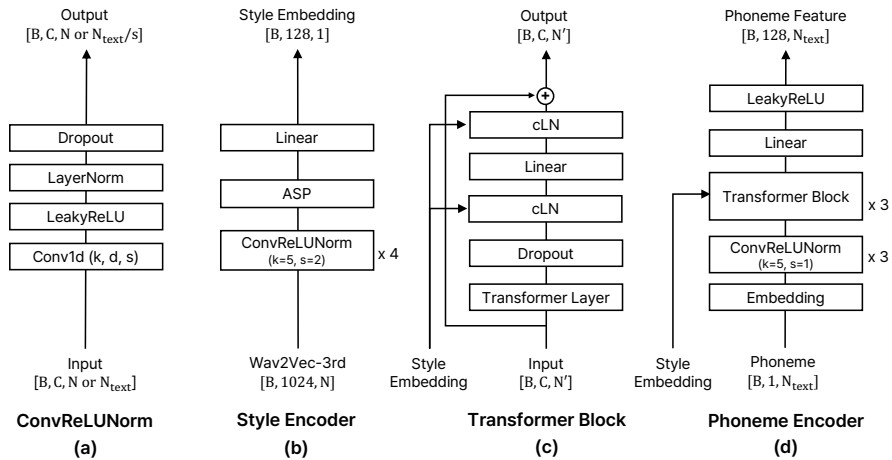

Figure 11: NANSY-TTS encoder architecture. k, d, and s of the Conv1d layer denotes kernel size, dilation, and stride, respectively.

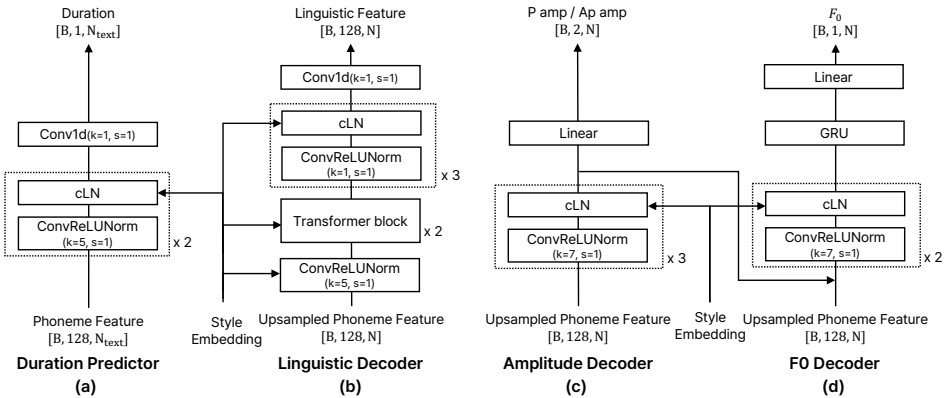

Figure 12: NANSY-TTS decoder architecture.

## B.3 DETAILED EVALUATION RESULT

**Training efficiency** To verify the efficiency of training time, we measured character error rate (CER (%)) at each training step. The experimental setting is the same as the setting for the full single-speaker dataset in section 4.2.2. As shown in Fig. 13, NANSY-TTS achieved low CER even though it was trained for only a few thousand steps. In comparison, the baseline model (Glow-TTS) showed relatively slower training convergence. In particular, NANSY-TTS trained for only 2k steps outperformed Glow-TTS trained for 200k steps. In addition, NANSY-TTS trained for 10k steps achieved lower CER than actual recordings. We speculate that this is because the NANSY++ analysis features are easier to model compared to the mel spectrogram, which is an entangled representation of amplitudes, pitch, linguistic information, and timbre.

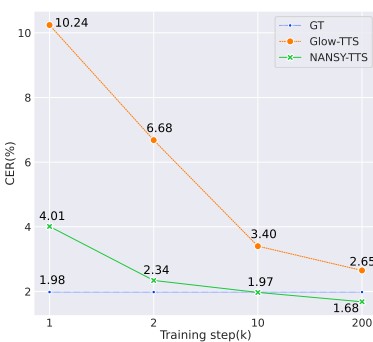

Figure 13: Fast training convergence.

**Zero-shot TTS results for each speaker** As shown in Table 5, NANSY-TTS achieved better MOS than actual recordings. This means that the raters felt that the speech generated by NANSY-

TTS were more natural than the actual recordings. Actual recordings often have diverse styles (e.g., speaking speed, pauses, intonation, etc.), which made the listener feel unnatural. In particular, it is noticeable in certain speakers of VCTK dataset, as shown in Table 8. The audio samples are provided in the demo page: tinyurl.com/8tnsy3uc.

Table 8: Zero-shot TTS results for each speaker. MOS(left) and SSIM(right)

| Speaker | Model Accent | GT | YourTTS | NANSY-TTS ($vctk$) | NANSY-TTS ($entire$) |
|---|---|---|---|---|---|
| p225 | English | 3.93 / 2.67 | 3.27 / 2.56 | 4.29 / 2.93 | 4.40 / 3.04 |
| p234 | Scottish | 3.89 / 2.47 | 3.91 / 2.64 | 4.31 / 2.73 | 4.31 / 3.27 |
| p238 | NorthernIrish | 4.22 / 3.11 | 3.58 / 2.44 | 4.31 / 2.24 | 4.36 / 2.02 |
| p245 | Irish | 4.29 / 2.44 | 3.11 / 2.07 | 4.36 / 2.42 | 4.11 / 2.42 |
| p248 | Indian | 3.60 / 3.13 | 2.93 / 1.78 | 4.33 / 1.80 | 4.36 / 2.29 |
| p261 | NorthernIrish | 4.09 / 2.98 | 2.89 / 1.91 | 4.20 / 2.84 | 4.33 / 3.29 |
| p294 | American | 4.16 / 3.00 | 3.84 / 2.20 | 4.16 / 3.20 | 3.93 / 2.49 |
| p302 | Canadian | 3.80 / 2.69 | 2.89 / 3.00 | 4.44 / 2.42 | 4.36 / 2.80 |
| p326 | Australian | 3.91 / 3.22 | 3.51 / 1.96 | 4.38 / 3.16 | 4.20 / 3.27 |
| p335 | NewZealand | 3.93 / 3.13 | 3.00 / 1.67 | 4.22 / 2.38 | 4.33 / 2.13 |
| p347 | SouthAfrican | 3.96 / 3.18 | 3.64 / 2.09 | 4.02 / 1.93 | 4.38 / 2.16 |

## C SINGING VOICE SYNTHESIS

### C.1 DETAILED ARCHITECTURE OF NANSY-SVS

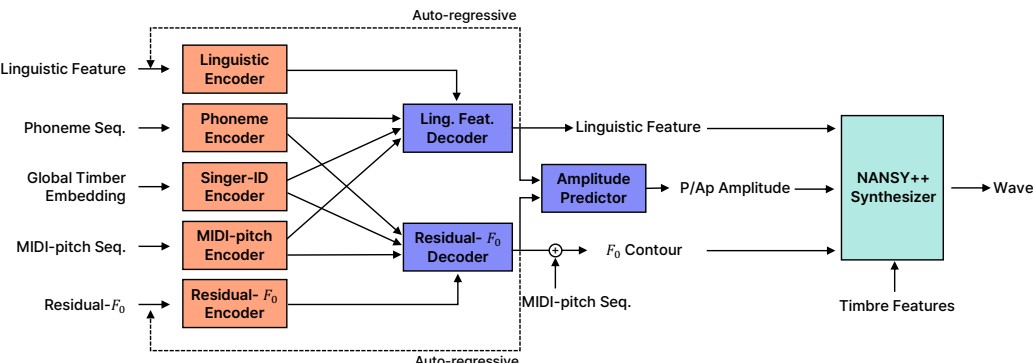

Figure 14: Overview of NANSY-SVS.

NANSY-SVS generates linguistic features and residual-$F_0$ in an autoregressive manner using a MIDI-pitch and phoneme sequence obtained from a musical score. In particular, to speed up an autoregressive inference, we first generate linguistic and residual-$F_0$ features downsampled to 1/4 time-resolution, and then upsample them with the original time-resolution through an neural upsampler module. The dimensions of the input and output tensors of the model are shown in the table 9.

As shown in Fig. 14, the NANSY-SVS model consists of five encoders, two decoder blocks, and an amplitude predictor. This section describes the detailed structure of each module in the order of encoder, decoder, and amplitude predictor.

**Encoder** Each encoder of NANSY-SVS is composed of a combination of PreConv and ConvGLU modules. PreConv consists of LeakyReLU activation and Dropout followed by a 1x1 1d convolutional layer. Unless otherwise stated, the dropout rate was fixed at 0.05. The ConvGLU module consists of Dropout, padding, 1d convolutional layer, and Gated Linear Unit. For the implementation of casual and non-causal characteristics, we used two different padding methods. For the

Table 9: The dimensions of the input and output tensors of NANSY-SVS. $B$ and $T$ denotes batch size and frame length, respectively.

| Name | Shape |
|---|---|
| **Input** | |
| midi-pitch sequence | [B, 1, T] |
| phoneme sequence | [B, 1, T] |
| linguisitic feature (downsampled) | [B, 128, T/4] |
| residual feature (downsampled) | [B, 241, T/4] |
| singer embedding | [B, 192, 1] |
| **Output** | |
| linguistic feature (downsampled) | [B, 128, T/4] |
| linguistic feature | [B, 128, T] |
| residual-$F_0$ (downsampled) | [B, 241, T/4] |
| residual-$F_0$ | [B, 241, T] |
| P amplitude | [B, 1, T] |
| Ap amplitude | [B, 1, T] |

causal module, padding was added to the front of the input sequence, and for the non-causal module, the total padding length was divided in half and added to both sides. In order to optionally condition the singer ID embedding, a conditional layer normalization layer Chen et al. (2020b) is added at the end of the module. Since the linguistic feature and residual-$F_0$ are generated in an autoregressive manner, they have to be causally encoded in the encoding step. Therefore, as shown in Figure 15-(c), Linguistic feature encoder and residual-$F_0$ encoder are designed to go through a total of 10 causal ConvGLU blocks after two preconv and a point-wise convolutional layer. To widen the receptive field, we increase the dilation of the ConvGLU block to a power of 3. Similarly, the encoder that processes the phoneme sequence and MIDI-pitch sequence is designed as shown in Figure 15-(d). Since phoneme and MIDI-pitch is categorical information, the embedding lookup table is added at the beginning of the module. In addition, a 1d convolutional layer with stride of 2 was added between ConvGLU modules to match the time-resolution of encoded feature with the downsampled target feature. We used the NANSY++ backbone timbre embedding as the singer ID embedding for NANSY-SVS. We assume that this embedding already contains enough information about the speaker, so the singer-ID encoder is designed as a simple fully-connected layer with ReLU activation.

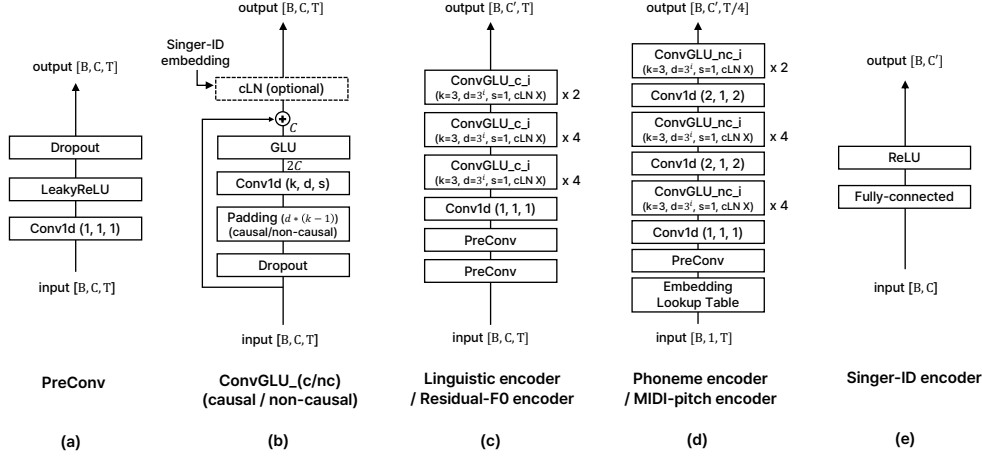

Figure 15: NANSY-SVS encoder architecture. $k$, $d$, and $s$ of the Conv1d layer denotes kernel size, dilation, and stride, respectively.

**Decoder**  Each decoder block of NANSY-SVS consists of casual and non-causal decoders and up-samplers as shown in Figure 16-(d). Both the casual and non-causal decoders (Figure 16-(a, b)) consist of 10 ConvGLU blocks and the last convolutional layer after PreConv module, and only the causality of ConvGLU is different. Singer-ID embedding is input as a condition to all ConvGLU blocks, and is reflected through the conditional layer norm layer. The upsampler (Figure 16-(c)) consists of the nearest upsample layers following the ConvGLU block to upsample the time-resolution of the input by 4 times. In the linguistic feature decoder block, the output of {Ling., MIDI-pitch, phoneme} encoder and singer-ID embedding are input to the causal decoder, and the output of the phoneme encoder and singer-ID embedding are input to the non-causal decoder. Similarly, in the residual-$F_0$ decoder block, the output of the {Residual-$F_0$, phoneme, MIDI-pitch} encoder and singer embedding are input to the causal decoder, and the output of the MIDI-pitch encoder and singer embedding are input to the non-causal decoder. Finally, the model is trained so that the sum of the causal decoder and the non-causal decoder equals to the downsampled target feature, and the sum of the outputs of the two upsamplers equals to the original time-resolution target feature.

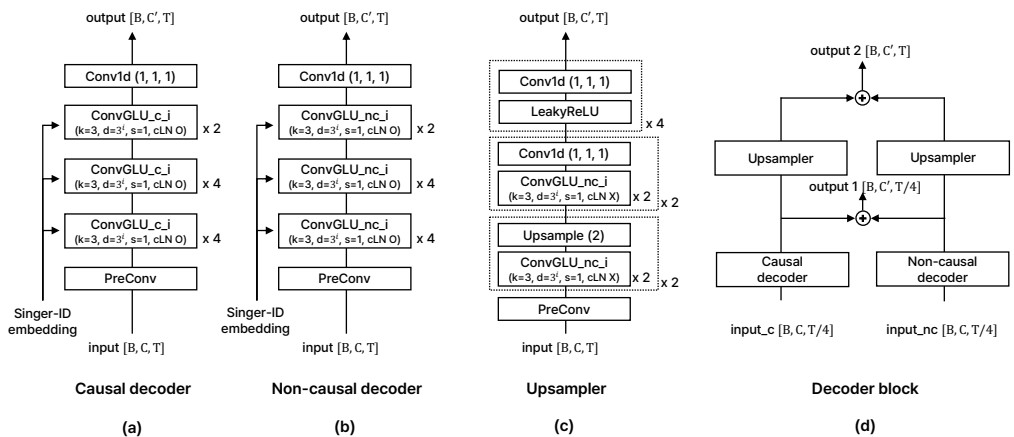

Figure 16: NANSY-SVS decoder architecture

**Amplitude Predictor**  The amplitude predictor has the same structure as Figure 8-(b) Non-causal decoder, except that singer embedding is not used as an conditional input. The predicted linguistic feature and residual-$F_0$ are concatenated and used as input, and P amplitude and Ap amplitude, one of the analysis features of the NANSY++ backbone, are predicted. For training stability, the amplitude predictor was trained independently using a stop gradient layer instead of training with the entire network.

### C.2  EVALUATION ON PITCH ESTIMATION AND VUV DECISION

For quantitative evaluation, we measured the $F_0$ RMSE and voiced/unvoiced decision error rate between the ground truth and the generated singing voice. To extract $F_0$ from singing voice, we used rapt (Talkin & Kleijn, 1995) pitch tracking algorithm. The evaluation results are shown in Table 10. When all the training data were used, both models achieved similar performance. However, as the volume of training size was set to be smaller, $F_0$ RMSE and VUV error rate of Diffsinger got degenerated significantly while those of NANSY-SVS did not. This demonstrates that NANSY-SVS faithfully reflects the input musical score condition even when trained with the small amount of data.

## D  VOICE DESIGNING

### D.1  DETAILED ARCHITECTURE OF NANSY-VOD

Fig. 17 presents the detailed structures of the flow networks used in NANSY-VOD. Each flow layer conducts three processing steps; (i) normalization of input with learnable parameters (ActNorm),

Table 10: $F_0$ and VUV difference between ground-truth and generated singing voice.

| MODEL | $F_0$ **RMSE (cent)** | **VUV error rate (%)** |
|---|---|---|
| BL / Ours $(80)$ | 233.69 / 91.18 | 12.93 / 5.67 |
| BL / Ours $(160)$ | 200.41 / 101.46 | 9.05 / 4.95 |
| BL / Ours $(320)$ | 125.97 / 94.96 | 7.39 / 5.35 |
| BL / Ours $(full)$ | 97.96 / 96.90 | 4.79 / 4.89 |

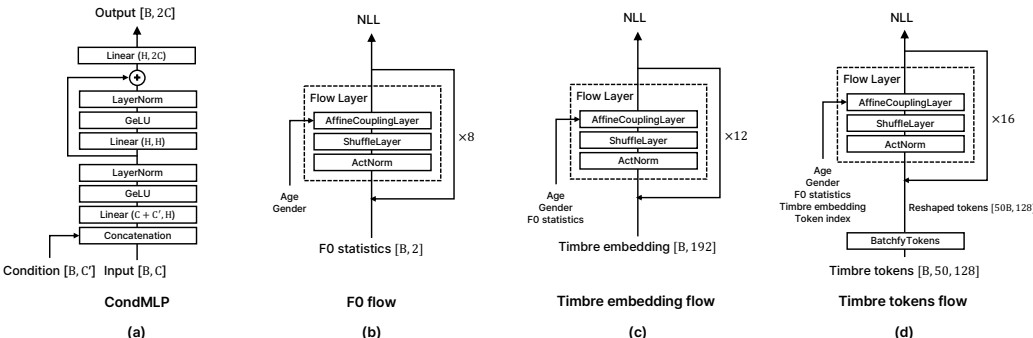

Figure 17: Flow architecture of NANSY-VOD.

(ii) shuffling input across channel dimension (ShuffleLayer), and (iii) applying an invertible bijective function (AffineCouplingLayer). ActNorm normalizes data by using the process $y = \frac{(x-\beta)}{\gamma}$ where $\beta$ and $\gamma$ are learnable parameters of the same dimension as the input dimension. $\beta$ and $\gamma$ are initialized such that $y$ has zero mean and unit variance given an initial training batch. ShuffleLayer swaps the first half of the channels with the other half. AffineCouplingLayer transforms the half of the input channels while keeping the other part as follows:

$$(\alpha, \omega) = \text{CondMLP}(x_b) \tag{2}$$
$$y_a = \alpha \odot x_a + \omega \tag{3}$$
$$y_b = x_b \tag{4}$$

where $x_a$ and $x_b$ denotes the first and second part of the input respectively.

NANSY-VOD consists of three flow networks; $F_0$ flow, timber embedding flow, and timbre tokens flow. $F_0$ flow employs 8 flow layers and uses age and gender as conditional variables. Similarly, time embedding flow is constructed of 12 flow layers conditioned by age, gender, and $F_0$ statistics. Timbre tokens flow consists of 16 flows layers with age, gender, $F_0$ statistics, global timbre embedding and token index as conditional terms. Timbre tokens are reshaped to form multiple individual data points, passed to 16 layers of timbre tokens flow, and processed independently. To distinguish timbre tokens, the corresponding token index is used as an additional conditional input.

### D.2 AGE CONTROL

We note that gender labels estimated by SPNet show a different trend depending on age. We annotated continuous gender labels of the 320 utterances using SPNet and report the result in Fig. 18. The gender labels of children under age of 10 are clustered around [-5, 5] while the other gender labels are scattered on both sides. From this, instead of adjusting an age value only, we devise an age control algorithm for NANSY-VOD taking into account the dependency between age and continuous gender. When the age value is adjusted from under 10 to over 10 years (i.e., from a child voice to a mature voice), the gender value is multiplied by 8. When the age value is edited from over 10 to under 10 years (i.e., from a mature voice to a child voice), the gender value is divided by 8. In other cases, we simply control the age value only. In our preliminary experiments, we empirically verified that this algorithm significantly improves the age editing performance of NANSY-VOD.

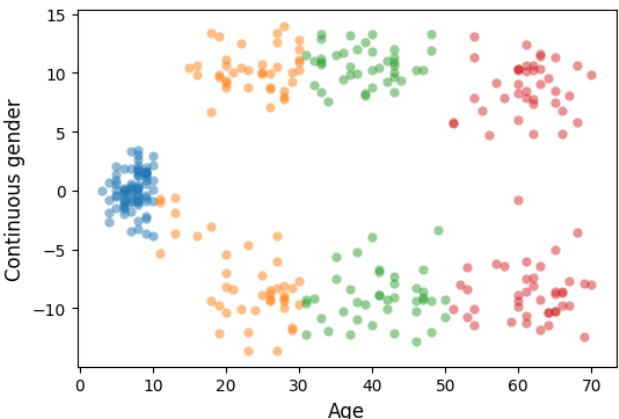

Figure 18: Estimated age and gender distribution on test set samples. The gender distribution appears to have bi-modal distribution for the people who are older than 10 years old, while children who are younger than 10 years old appears to have uni-modal distribution around 0.

## D.3 INFERENCE OVERVIEW

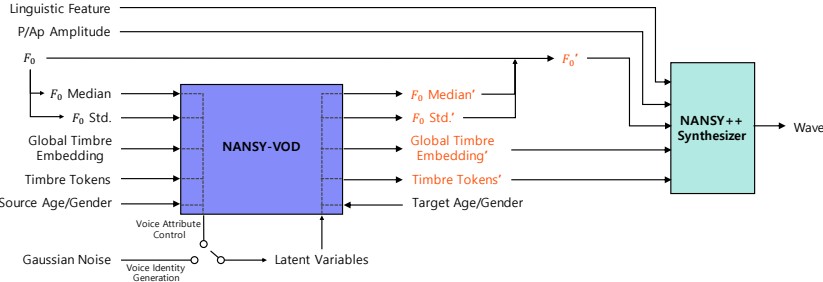

Figure 19: Overview of inference procedure of NANSY-VOD.

Fig. 19 demonstrates the inference procedure of NANSY-VOD. To edit the voice attributes of source audio, NANSY-VOD transforms F0 statistics, global timbre embedding, and timbre tokens into latent variables, and converts them back to analysis features. Note that NANSY-VOD uses a source speaker's age and gender for the forward process, and the target age and gender for the inverse process. To generate a new voice identity, latent variables are sampled from the normal distribution and transformed into analysis features according to the target age and gender.

## D.4 SPNET

SPNet is a neural network for estimating speech parameters such as age and gender given speech. SPNet employs the same architecture as ECAPA-TDNN (Desplanques et al., 2020) and stacks a projection layer on top of it. The last layer projects 192 dimension to 2 dimension, and each output is used to estimate age and gender. Mean absolute error and soft margin loss are employed as an objective function for age and gender estimation, respectively. AI-Hub conversation datasets were used for both training and evaluation. We initialized SPNet with the pretrained ECAPA-TDNN weights and trained it using the AdamW optimizer (Loshchilov & Hutter, 2017) of learning rate $2 \times 10^{-4}$ for about 33K iterations. Each audio data was randomly cropped to one second and used to construct a mini batch of size 256. At test time, we sampled 320 utterances from various age and gender groups, and estimated speakers' age and gender using SPNet. The average absolute error of age estimation was 4.095 and the average accuracy of gender estimation was 93.75%. Since voices of pre-pubescent children may not be gender distinct, we also evaluated gender estimates for people over the age of 10. In this case, the average accuracy recorded 99.17%.

