# OpenReview forum: "NANSY++: Unified Voice Synthesis with Neural Analysis and Synthesis"
_ICLR.cc/2023/Conference — ICLR 2023 poster_

### Official Review · Reviewer_vW43 · 2022-10-22

**Confidence:** 4
**Correctness:** 2
**Technical Novelty And Significance:** 2
**Empirical Novelty And Significance:** 3
**Recommendation:** 5

**Clarity, Quality, Novelty And Reproducibility:**

The paper's writing can be improved by clearly stating assumptions -- revisiting this assumption in the experimental sections and verifying them. Stating what is claimed, vs which choices are made to encourage learning specific kind of features, even when they are not guaranteed to happen, will be useful.

The paper re-uses components from other papers and cites them. Paper has limited novelty in terms of methods proposed but seems to be a useful well-engineered system.

**Strength And Weaknesses:**

- Overall:
    - Authors claim about disentangled representation learning of f0, timbre, linguistic information might be misleading. I wonder if authors will consider changing the wording to showcase what design choices they make that promote learning disentangled representation, and they verify experimentally that some of it is achieved.
    - Experimental results on downstream tasks seems very good. The impact of the work can be maximized by making sure that the baseline performance is replicated from original work e.g. baseline TTS on full data, seems to be very far from real data MOS, which does not reflect the SOTA on TTS tasks.

Strengths:
- Extensive evaluation
- Impressive Results

Weakness:
- Not so great scientific rigor
- Not so-great choice of baselines
- Potentially misleading claims or just confusing to understand what is claimed vs what is expected.

**Summary Of The Paper:**

- Authors propose a unified framework to learn many voice synthesis tasks. The unified framework includes task-independent self-supervised training of a backbone network, which does not require any labelled data, and then task specific training with labelled data.
- Authors claim that this unified approach offers advantages such as controllability of synthesized speech, data efficiency for downstream tasks, and fast training convergence — all this without sacrificing the quality of synthesized speech.
- Authors assume that the most of the voice synthesis tasks can be defined synthesizing and controlling our aspects of voice, that is, pitch, amplitude, linguistic, and timbre.
- For estimating pitch, in an unsupervised fashion, authors use the property of CQT input representation, and the transforming the representation appropriately to get a known shift in the fundamental frequency value. While this will bias learning of fundamental frequency like value, it is not clear how authors obtain absolute fundamental frequency. This method is inspired by “SPICE: Self-supervised Pitch Estimation”.
- For getting linguistic features, authors propose to use a contrastive learning approach where they modify the signals in a way to keep the linguistic information intact, while changing other properties of the signal — and use this property learn features which maximizes similarity if the source audio (i.e. before transformations) is the same i.e positive samples in the contrastive learning approach are two different transformations of the same signal and negative samples are the transformations of two different signals. Negative samples clearly have different linguistic information, but it is not clear that they will share the same non-linguistic features — maybe that is not necessary for this approach to work. Authors extract wav2vec features from transformed signal, which is transformed via another learnable network. I am curious about the importance of this contrastive learning approach vs directly using wav2vec features.
- Authors propose learnable vector encoding of input signal — one global vector and then time varying tokens — to represent timbre. It is not clear why these learnable encodings will necessarily encode timbre, and will necessarily be disentangled from pitch and linguistic representation.
- Authors propose to use Parallel WaveGan trained via reconstruction loss  on mel-spectrogram, and adversarial and feature matching loss to train the synthesizer network.
- Experiments:
    - Authors shows that replacing fundamental frequency extracted from the analysis network by the fundamental frequency extracted from four other approaches, and show that the synthezised audio using their synthesizer is assessed to match the original signal before doing the f0 replacement. It is not clear why this implies that their analyser learns better f0 — it is totally possible and likely that even if their analyser is not learning the correct f0, but since the representation is trained end-to-end along with synthesizer, reconstructions with this representation are closer to original, and using a more correct (or even a different f0) will throw the model off.
    - Authors show better reconstruction quality than a mel2wav architecture namely HiFiGAN. Since the input representations are different, specifically authors choice of input is much more rich than mel-spectrogram used in mel2wav setup. It is not clear why having better reconstruction is a good metric to measure in this case.
    - Authors show clear improvement in zero-shot voice conversion, when compared with the baseline.
    - Authors show better performance than the baseline on TTS and Singing voice synthesis. The baseline for full data seem far from real audio, which probably points to the baseline being weaker than SOTA TTS.
    - Authors show good results on Voice Designing.

**Summary Of The Review:**

Overall, the authors should reconsider claims about disentangled representation learning. Should consider doing ablations for various choices, and improve clarity of the writing.

---

> ### Author Response · Authors · 2022-11-08
> **Response to Reviewer vW43 (1/3)**
>
> We appreciate reviewer vW43's effort for reviewing our paper.
> Below are the responses to your concerns and questions.
>
>
> \
> \
> **Review**: For estimating pitch, in an unsupervised fashion, authors use the property of CQT input representation, and the transforming the representation appropriately to get a known shift in the fundamental frequency value. While this will bias learning of fundamental frequency like value, it is not clear how authors obtain absolute fundamental frequency. This method is inspired by “SPICE: Self-supervised Pitch Estimation”.
>
> **Answer** : We understand the confusion. Unlike the SPICE approach, however, our self-supervised pitch estimation method differs in that it is trained within the analysis-synthesis reconstruction loop. Meaning that the pitch estimator must estimate pitch-related value using a sequence of scalars that contributes to reconstruction of an input signal, which is why it turns out to be a fundamental frequency as a result.
> We have attached the output of the pitch estimator in the Appendix A.2: Qualitative Results on $F_0$ estimation section, along with the log frequency spectrogram in order to convince reviewers and readers that it indeed excatly follows the trajectory of the fundamental frequency part.
>
>
>
> \
> \
> **Review**: Negative samples clearly have different linguistic information, but it is not clear that they will share the same non-linguistic features — maybe that is not necessary for this approach to work. Authors extract wav2vec features from transformed signal, which is transformed via another learnable network. I am curious about the importance of this contrastive learning approach vs directly using wav2vec features.
>
> **Answer** : Thank you for pointing out the importance of the contrastive loss.
> The mid-level features of wav2vec have rich information that is enough to fully reconstruct the original signal, meaning that it’s not a disentangled representation for pure linguistic information (this was shown by [1] and [2] before, and some of our experiments too.). Which is why we believe more layers on top of the wav2vec network is required.
> Futhermore, we found that only using the information perturbation method does not make the linguistic encoder to fully encode the linguistically disentangled representation, but it’s actually the decoder that selectively takes the information from the encoded representation (this was also stated in [2] Related work section on Consistency Learning).
> We believe this calls for the need of the contrastive method to accelerate disentangeld representation learning. The effectiveness of such contrastive losses for classification tasks was also demonstrated by recent works such as [1] and [3].
> To further address the concern of the reviewer, we have conducted additional experiments by comparing phoneme classification accuracies of mel spectrogram, wav2vec-15th-feature, and NANSY++ linguistic feature. We trained frame-level phoneme classifier using 13 hours of singing voice dataset manually annotated with time-aligned phoneme labels. We used 8-layer convGLU stack with first and last 1x1 convolutional layer for phoneme classifier. We trained and evaluated three classifiers using mel spectrogram, wav2vec-15th-feature, and NANSY++ linguistic feature as input representation. The frame-level phoneme classification accuracies on the test set was 66.39% (mel), 68.41% (wav2vec), and 70.10% (NANSY++ linguistic feature), respectively. From this, we can see that the NANSY++ lingusitic feature was trained with a more suitable representation to distinguish phonemes.
>
>
> [1]: Qian, Kaizhi, et al. "ContentVec: An improved self-supervised speech representation by disentangling speakers." *International Conference on Machine Learning*
> . PMLR, 2022.
>
> [2]: Choi, Hyeong-Seok, et al. "Neural analysis and synthesis: Reconstructing speech from self-supervised representations." *Advances in Neural Information Processing Systems*
>  34 (2021): 16251-16265.
>
> [3]: Huang, Wenyong, et al. "Spiral: Self-supervised perturbation-invariant representation learning for speech pre-training."  *International Conference on Learning Representations*
> , 2022.
>
> \
> \
> **Review**: Authors propose learnable vector encoding of input signal — one global vector and then time varying tokens — to represent timbre. It is not clear why these learnable encodings will necessarily encode timbre, and will necessarily be disentangled from pitch and linguistic representation.
>
> **Answer**: Thank you for pointing this out. We agree that there could be confusion why the timbre encoder does not captures the content-related features such as pitch or linguistic information. This is because the timbre encoder extracts information into a fixed number of codes regardless of the length of the target waveform. In addition, the encoding process is permutation invariant and does not care about the sequential information using cross-attention between the encoded information and trainable query vectors.

---

> > ### Author Response · Authors · 2022-11-08
> > **Response to Reviewer vW43 (2/3)**
> >
> > **Review**: Authors propose learnable vector encoding of input signal — one global vector and then time varying tokens — to represent timbre. It is not clear why these learnable encodings will necessarily encode timbre, and will necessarily be disentangled from pitch and linguistic representation.
> >
> > **Answer**: Thank you for pointing this out. We agree that there could be confusion why the timbre encoder does not captures the content-related features such as pitch or linguistic information. This is because the timbre encoder extracts information into a fixed number of codes regardless of the length of the target waveform. In addition, the encoding process is permutation invariant and does not care about the sequential information using cross-attention between the encoded information and trainable query vectors.
> >
> > \
> > \
> > **Review**: Authors shows that replacing fundamental frequency extracted from the analysis network by the fundamental frequency extracted from four other approaches, and show that the synthezised audio using their synthesizer is assessed to match the original signal before doing the f0 replacement. It is not clear why this implies that their analyser learns better f0 — it is totally possible and likely that even if their analyser is not learning the correct f0, but since the representation is trained end-to-end along with synthesizer, reconstructions with this representation are closer to original, and using a more correct (or even a different f0) will throw the model off.
> >
> > **Answer**: We agree that there could be such concerns. This is related to your question above regarding the absolute F0 estimation, so we hope that your concern on the pitch information is already somehow alleivated. In addition, if what you are saying is true, the model would have produced weird signal when using F0 from other models. But in our experiments the F0 sequence extracted from other off-the-shelf F0 estimators worked just fine (’we have newly added the reconstruced signal in the demo page’) and can be reconstructed with the exact same pitch, especially when the F0 is extracted well from a clean voice signal.
> > We hope the newly added section, Appendix A.2: Qualitative Results on $F_0$ estimation section, can somehow address your concerns.
> >
> >
> > \
> > \
> > **Review**: Authors show better reconstruction quality than a mel2wav architecture namely HiFiGAN. Since the input representations are different, specifically authors choice of input is much more rich than mel-spectrogram used in mel2wav setup. It is not clear why having better reconstruction is a good metric to measure in this case.
> >
> > **Answer** : As stated in 3.2 Reconstruction section, we conducted this experiment in order to validate that the backbone synthesizer can be used as a synthesis module for various applications. We wanted to measure the maximum quality of NANSY++ when used as a waveform synthesis module for other applications. Because we have conducted various experiments on actual applications using HiFi-GAN, we do not think this is really a problem. In addition, it is not necessarily true that mel spectrogram contains less information than the NANSY++ features. NANSY++ features, especially the linguistic information, were encouraged to be disentangled, therefore, there is a possibility that some information was lost during the training, whereas mel spectrogram was extracted right away from the raw signal.
> > Finally, one of the reasons why we conducted this experiment is to point out that using mel spectrogram as a mid-level feature is not really an optimal solution for better vocoding.

---

> > > ### Author Response · Authors · 2022-11-08
> > > **Response to Reviewer vW43 (3/3)**
> > >
> > > **Review** : Experimental results on downstream tasks seems very good. ***The impact of the work can be maximized by making sure that the baseline performance is replicated from original work. e.g. baseline TTS on full data, seems to be very far from real data MOS, which does not reflect the SOTA on TTS tasks.
> > >
> > > **Answer** : Thank you for pointing out the performance of baseline models. I think there are some misunderstandings here because we have only used the official implementations of the original authors (GLOW-TTS ([https://github.com/jaywalnut310/glow-tts](https://github.com/jaywalnut310/glow-tts)), HiFi-GAN ([https://github.com/jik876/hifi-gan](https://github.com/jik876/hifi-gan)), YourTTS ([https://github.com/Edresson/YourTTS](https://github.com/Edresson/YourTTS)), DiffSinger([https://github.com/MoonInTheRiver/DiffSinger](https://github.com/MoonInTheRiver/DiffSinger))). In addition, as far as we are concerned YourTTS is the most recent paper on zero-shot TTS with the official implementation of the original authors, and it shows competitive performance on zero-shot TTS performance because it’s an extension of the SOTA TTS model VITS. As for the singing voice synthesis part, we chose the official implementation of DiffSinger as our baseline model because not only it is the recent work, but also because it shows the most competitive performance on singing voice synthesis task among the other open sourced singing voice synthesis models. Please notice also that we have used the exact same data configurations for training baseline models and our model so that the experiments are all fair.
> > >
> > > \
> > > \
> > > **Review** : The paper's writing can be improved by clearly stating assumptions -- revisiting this assumption in the experimental sections and verifying them. Stating what is claimed, vs which choices are made to encourage learning specific kind of features, even when they are not guaranteed to happen, will be useful.
> > >
> > > **Answer** : Thank you for your suggestion for making our paper more solid. We believe the most important experiments of our paper is the data efficiency part, which we have focused most of our time to evaluate. The assumption was “by exploiting self-supervised representation learning strategies for downstream generation tasks, we can encourage each task to be more data efficient, while not losing modularity and the loss of synthesis quality“.
> > > We have modified the manuscript by stating the assumption in the introduction.

---

### Official Review · Reviewer_Hzqp · 2022-10-23

**Confidence:** 4
**Correctness:** 3
**Technical Novelty And Significance:** 2
**Empirical Novelty And Significance:** 3
**Recommendation:** 5

**Clarity, Quality, Novelty And Reproducibility:**

Clarity
- The paper is easy to read in general. But, all details are located in the appendix section. I am not sure this type of writing structure is appropriate as a conference paper.
- Since the framework include a lot of components, the authors should clarify which part is novel and how they affect the model performance.

Quality
- The model quality was well evaluated on the backbone experiment and each application task with MOS, SSIM, CER and more.
- F0 estimation was evaluated only with a listening test for the reconstruction quality. It could be also evaluated in terms of F0 estimation accuracy.

Novelty
- While a lot of components are borrowed from previous work, there are also some new ideas such as time-varying timbre embeddings. But, the novel part was not clearly validated. For example, the time-varying timbre embeddings could be evaluated in the reconstruction experiment (section 3.2) as an ablation study.

Reproducibility
- The model architecture is well delineated in the appendix section.
- However, the detail of the training data is missing. Information on gender, age, language, professionality (they include voice actors or professional singers, etc.) would be helpful to understand the model training.

Minor comments
- (page 2) the sinusoidal waveform equation ( x[t] = Ap[t] sin (...)) has the summation inside the sine function. Is it correct? The summation should be moved outside the sine function?

- (page 3) In equation (1), "cossim" could be replaced with a short symbol such as "d". In page 9, they use the symbol "d" to denote the cosine distance. Please make them consistent.

- (page 4) what is "simple spectrogram"?  Is it the spectrogram with the linear frequency scale?



**Details Of Ethics Concerns:**

The proposed model can be used for voice conversion in the zero-shot setting, which can be potentially used as a "deep-fake" technology for voice. This is actually a general problem of voice synthesis and conversion. The authors acknowledge it and they also state counteracting technologies such as anti-spoofing.



**Strength And Weaknesses:**

Strengths
- The proposed framework is highly modular and flexible, being plugged in various voice synthesis tasks
- The framework is interpretable as it is designed to be similar to DSP-based parametric vocoders.
- They achieve impressive results in all tasks, outperforming compared models.

Weakness
- The paper includes too much content. Due to the space issue, all details are moved to the appendix.
- Although the proposed framework is well-structured, it is extended from the authors' previous work (NANCY) and many new components are adopted from other previous work (self-supervised pitch estimation, linguistic representation, parallel WaveGAN, and so on).
- While the framework achieves better performances than previous work in many tasks, the experiment condition is not completely fair because the NANCY framework was pretrained with a large-scale dataset (10,571 hours from 6,176 speakers and 624 singers).
- The time-varying timbre embedding is a great idea but it was not validated through an ablation study.


**Summary Of The Paper:**

This paper presents a unified framework for human voice synthesis based on neural analysis and synthesis modules. The analysis part extracts disentangled and controllable voice features based on domain knowledge (pitch, periodicity/aperiodicity and timbre). Each of them is trained independently with a dedicated neural analysis module. The timbre analysis module divided the feature into a global timbre embedding vector and time-varying timbre tokens.  The synthesis part take the voice features and generates waveforms with frame-level and sample-level synthesis modules. The framework is evaluated in terms of F0 estimation and reconstruction error as a vocoder, showing superior results.  The framework is applied to four voice synthesis tasks including voice conversion, TTS, singing voice synthesis, and voice design, along with additional encoder modules. In each task, the proposed framework outperform previous state-of-the-arts.




**Summary Of The Review:**

This paper proposes a flexible voice synthesis framework with high-performance. The framework include robust F0 estimation and high-quality reconstruction as a vocoder.  The flexible structure was applied to voice conversion, TTS, singing voice synthesis and new voice design. I believe that the framework is highly well designed and engineered.

One main issue of this paper is the writing structure. The main body is a summary of long manuscripts on "a large system" and all details are moved to the appendix section. I feel like it would be better for the authors to submit this paper to a journal which allows many pages.

In addition, the novel part such as time-varying timbre embedding was not thoroughly validated, which might be probably due to the space issue.

Lastly, the high-performance of the framework is attributed to not only the model architecture but also the large-scale training data. This should be clearly indicated when compared to other models.

---

> ### Author Response · Authors · 2022-11-08
> **Response to Reviewer Hzqp**
>
> We appreciate reviewer Hzqp's effort for reviewing our paper.
> Below are the responses to your concerns and questions.
>
> \
> \
> **Reviewer**: While the framework achieves better performances than previous work in many tasks, the experiment condition is not completely fair because the NANCY framework was pretrained with a large-scale dataset (10,571 hours from 6,176 speakers and 624 singers)
>
> **Answer**: With all due respect, for fair comparisions, we actually matched the dataset configurations for baseline models for every experiment, and it is written on our manuscript. 1. We re-trained HiFi-GAN on the same dataset using the official implementation. 2. We trained GLOW-TTS and NANSY-TTS module both on the same HiFi-TTS dataset. 3. We didn’t retrain Your-TTS but we retrained NANSY++ using the same dataset configuration following the description in official Your-TTS implementation. 4. We trained DiffSinger and NANSY-SVS module on the same OpenCpop dataset.
>
> \
> \
> **Reviewer**: The time-varying timbre embedding is a great idea but it was not validated through an ablation study.***
>
> **Answer**: Thank you for pointing this out. It would have been better if we could have conducted experiments for each analysis module but we did not have enough resources to train NANSY++ backbone for all of them. In our initial experiments, however, we have indeed found significant improvement on the zero-shot voice conversion quality, although it is conducted through an internal listening test.
>
> \
> \
> **Reviewer**: Since the framework include a lot of components, the authors should clarify which part is novel and how they affect the model performance.
>
> **Answer**: Thank you for your suggestion for making the paper more concise and clear for readers. We understand that it could have been better if we had done more exhaustive ablation studies on each part of the model, but this really requires lots of computation resources for our organization. We believe the most significant novelty of this study is achieved not by proposing completly new modules (although there are some contributions on each of them), but rather by integrating them into a single analysis-synthesis training loop. This really opens up a new training scheme for voice synthesis tasks because there has been no such studies that gurantess several advantages such as modularity, controllability, wide range of applicability and high-quality synthesis at the same time. Therefore, we think the real contribution of our work comes from the systemical contribution of integrating them into a single framework.
>
>
>
> \
> \
> **Reviewer**: F0 estimation was evaluated only with a listening test for the reconstruction quality. It could be also evaluated in terms of F0 estimation accuracy.
>
> **Answer**: We agree on this concern. However, it’s actually hard to estimate the F0 estimation accuracy on voice signals especially because it is almost impossible to obtain the ground truth F0 sequence from the signals. This is why we had to resort on listening test in the end.
>
>
> \
> \
> **Reviewer**: However, the detail of the training data is missing. Information on gender, age, language, professionality (they include voice actors or professional singers, etc.) would be helpful to understand the model training.
>
> **Answer**: We understand that it could be helpful for the reproducibility if there were such information. However, because we are exploiting the self-supervised training learning methods here, we actually do not have such high-level descriptions for most of our training samples.
>
>
> \
> \
> **Reviewer**: (page 2) the sinusoidal waveform equation ( x[t] = Ap[t] sin (...)) has the summation inside the sine function. Is it correct? The summation should be moved outside the sine function?
>
> **Answer**: The summation inside the sine function has no problem as it is the cumulative sum process of instantaneous frequency to a phase signal. However, we think the index of F0 could be confusing, so we have modified this in our new manuscript.
>
>
> \
> \
> **Reviewer**: (page 3) In equation (1), "cossim" could be replaced with a short symbol such as "d". In page 9, they use the symbol "d" to denote the cosine distance. Please make them consistent.***
>
> **Answer**: Thank you for pointing this out, we have changed the notation following your suggestion.
>
>
> \
> \
> **Reviewer**: (page 4) what is "simple spectrogram"? Is it the spectrogram with the linear frequency scale?
>
> **Answer**: Yes, it means the spectrogram with the linear frequency scale. We have changed it into “linear frequency scale spectrogram” in our new manuscript.

---

### Official Review · Reviewer_D823 · 2022-10-24

**Confidence:** 3
**Correctness:** 4
**Technical Novelty And Significance:** 3
**Empirical Novelty And Significance:** 3
**Recommendation:** 8

**Clarity, Quality, Novelty And Reproducibility:**

See above, but generally the paper is clearly written and high quality experiments. One minor comment is that I would recommend avoiding the acronym "VD" for "voice designing" as there is already an unpleasant meaning to this acronym.

In terms of novelty, the approach is based upon an earlier work, NANSY (Choi et al, 2021b), but adds the trainable transformations in the analysis block, which enables the end-to-end training. This paper also applies it to a number of new tasks, which were not necessarily easy to target in the earlier version.

On reproducibility, the paper provides details of all of the architectures as well as hyperparameters in the appendix. Datasets are publicly available. Code is not released because of concerns about the potential of the model to be misused.

**Details Of Ethics Concerns:**

As with any low-data voice conversion approach, there is the possibility of the method being used to create recordings of statements that speakers never made. This is acknowledged by the authors in their ethics statement and they have withheld the release of their code from the public, but promise to make it available to "identified and authorized users" who request it. I think this is acceptable, but that it should be mentioned.

**Strength And Weaknesses:**

Strengths:
* Well executed experiments: the experiments are all clearly described, well performed, and test key claims of the paper
* Approach works well. The results of those experiments show that the proposed method does provide superior performance to the various baseline approaches.
* Model provides general utility. As demonstrated by the variety of experiments conducted, the proposed model is useful in a large number of situations without requiring a large amount of data or a large amount of time to retrain. For example, for TTS, between 30 minutes and 30 hours of data, the model improves in MOS from 3.64 to 4.07 and in CER from 2.20 to 1.68%. These are meaningful improvements, but the model trained on 30 minutes is still performing quite well and fairly close to the model trained on 30 hours.
* General clarity of presentation. The paper is generally well presented, except for the issue of certain claims being slightly over stated mentioned below under weaknesses.

Weaknesses:
* The captions of the figures and tables are too brief and should be expanded so that figures are clearly interpretable by themselves without referring to the text.
* Certain claims are slightly overstated, although these claims are also not especially necessary for the work to have impact. In particular, I am referring to claims that the model is disentangling various voice characteristics and that it is self-supervised. While both of these may be technically true, it is thanks to careful inductive biases baked into the model architecture. Thus it is not especially surprising that the model has these characteristics, but it's not like they just popped out of the data.

**Summary Of The Paper:**

This paper describes a learned analyzer and synthesizer for speech that can be trained end-to-end, and demonstrates their utility in several speech and singing synthesis tasks: voice conversion to match a target individual, voice conversion to match target characteristics, text to speech, and singing voice synthesis. In listening tests for the targeted voice conversion, NANSY++ exceeded YourTTS  by 0.56 MOS points (from 1-5 range) in quality and 0.38 points (1-5 range) in speaker similarity. For TTS, compared with Glow-TTS, the proposed approach performs better in quality and character error rate of a pre-trained recognizer at various amounts of training data from a single speaker (5 mins, 10 mins, 30 mins, and 30 hours). For singing voice synthesis, the proposed approach out performed diffsinger in terms of subjective quality with various amounts of training data. And for voice conversion to match target characteristics, the proposed approach was able to generate 320 speakers that achieved an objective voice diversity similar to that of 240 actual different speakers.

**Summary Of The Review:**

End-to-end trainable voice analysis-synthesis pipeline that cleverly separates various voice characteristics through relative contrasts. Experiments show that it is useful and data efficient in several different tasks. Minor issues with clarity of captions and with certain claims being slightly overstated.

---

> ### Author Response · Authors · 2022-11-08
> **Response to Reviewer D823**
>
> We thank reviewer D823 for the extensive review.
> We also thank you for appreciating the novelties made by the end-to-end backbone training.
> Below are the responses to some of your concerns.
>
> \
> \
> **Reviewer**: The captions of the figures and tables are too brief and should be expanded so that figures are clearly interpretable by themselves without referring to the text.
>
> **Answer**: Thank you for your suggestion to make our paper more clear. We have changed the captions following your suggestion, but it was actually hard to make it completely self-contained because of the page limit.
>
> \
> \
> **Reviewer**: Certain claims are slightly overstated, although these claims are also not especially necessary for the work to have impact. In particular, I am referring to claims that the model is disentangling various voice characteristics and that it is self-supervised. While both of these may be technically true, it is thanks to careful inductive biases baked into the model architecture. Thus it is not especially surprising that the model has these characteristics, but it's not like they just popped out of the data.
>
> **Answer**: We agree that the disentanglement is not achieved solely by the self-supervised training method, but also the inductive bias from the generator architecture (e.g., using sinusoidal transformation of fundamental frequency as an input).
>
> \
> \
> **Reviewer**: See above, but generally the paper is clearly written and high quality experiments. One minor comment is that I would recommend avoiding the acronym "VD" for "voice designing" as there is already an unpleasant meaning to this acronym.
>
> **Answer**: Thank you for your detailed review. We have changed it into VOD in our new manuscript.

---

### Official Review · Reviewer_U62B · 2022-10-24

**Confidence:** 5
**Correctness:** 3
**Technical Novelty And Significance:** 3
**Empirical Novelty And Significance:** Not applicable
**Recommendation:** 8

**Clarity, Quality, Novelty And Reproducibility:**

The paper is well-written and technically solid. However, ablation study is largely missing.

**Strength And Weaknesses:**

It's impressive that NANSY++ can use one unified framework to handle four different speech tasks while achieving good performance in audio quality, data efficiency and controllability. The framework is well designed in a modularized way and can disentangle various speech representations. One drawback of this paper, however, is limited ablation study. Since the proposed NANSY++ framework requires a variety of modules and augmentations, it's hard to know what are the core factors that contribute to the model performance.

Several detailed questions:

In section 2.2, for the linguistic feature disentanglement, what's the exact equations of breathiness perturbation?

In section 2.3, are the global and time-varying timbre embeddings also trained in a self-supervised way or trained jointly with the waveform reconstruction? If they are jointly trained, how to guarantee the time-varying timbre embeddings don't contain other information such as pitch and linguistic?

In section 4.2.2, what's the problem definition here for zero-shot TTS? Does the mean the model trained on TTS task can be adapted to a new dataset or the pre-trained backbone model can be used directly for TTS task?

**Summary Of The Paper:**

This paper proposed NANSY++, a unified framework of synthesizing and manipulating voice signals from analysis features. The backbone network of NANSY++ is trained in a self-supervised way. After the backbone network is pre-trained, different voice applications can be adapted by modeling the analysis features required for each task. Experiments show that the proposed NANSY++ framework has the advantages of controllability, data efficiency, fast convergence, and high quality synthesis.

**Summary Of The Review:**

The proposed unified voice synthesis framework NANSY++ achieved impressive performance and can serve as a general speech pre-processing and pre-training pipeline.


=================

Updates after reading the author's rebuttal: the author addressed my concerns well, so I updated my rating to 8.

---

> ### Author Response · Authors · 2022-11-08
> **Response to Reviewer U62B (1/2)**
>
> We thank reviewer U62B for making efforts to review our paper.
> We also thank you for appreciating the strength of the unified framework.
> Below are the responses to your concerns.
>
> \
> \
> **Reviewer**: One drawback of this paper, however, is limited ablation study. Since the proposed NANSY++ framework requires a variety of modules and augmentations, it's hard to know what are the core factors that contribute to the model performance.
>
> **Answer**: We agree that the paper could seem like that. One of the reasons why it’s not easy to check every root of the performance is because there are multiple performance factors and most of them are due to the compounding effects from the end-to-end training. To address the concern, we have tried to organize the reasons where the performance boost comes from below.
>
> 1. Fundamental frequency estimation: Although ground truth fundamental frequency sequence cannot be accurately obatined from real-world voice signals, the proposed end-to-end self-supervised pitch training strategy can boost the performance by scaling up only the amount of waveform signals without paired annotations. We have empirically validated this in the experiment section. For example, the baseline model such as Crepe trained using synthetic dataset with paired F0 annotations peformed worse than our model, showing that our model shows better performance on real-world dataset.
>
> 2. Modularity & Fast training convergence: Because we have trained each feature representation to be disentangled, we believe it is easier for models to estimate the features compared to mel spectrogram that has relatively complex structure with entangled information. Furthermore, because NANSY++ synthesizer already deals with the timbre part, the application modules does not have to deal with the timbre synthesis, which alleivates the burden that happens when trying to estimate mel spectrogram.
> If we had to conduct experiments on analyzing this part, I’m not sure which components should we change to experiment on this. I don’t think intentionally making entangled representation by not using information perturbation would necesarily show something else because the training just fails without it (and it’s been already shown in the previous work [2]).
>
> 3. High synthesis quality: The high synthesis quality mostly stems from reducing glitches from singing voice. This is achieved by the sinusoidal input to the generator. We have found this in our initial experiments that models using sinusoids as input (e.g., neural source filter (NSF)) does not exhbit such glitches although the output synthesis quality is somewhat robotic. The glitch usually happens near the harmonic part (which is directly related to pitch) and providing explicit sinusoidal signal to the generator can alleviate this issue directly. The major difference of our model, however, is that we do not use external fundamental frequency estimator but rather uses the pitch analysis module trained in an end-to-end manner for a robust pitch estimation. In addition, end-to-end adversarial training helps improve the synthesis quality. Although we can try conducting additional experiments by comparing with normal parallel wavegan without sinusoidal input, we don’t think this is really necessary as it is already known to exhbit such glitches.
>
> [1] Wang, Xin, Shinji Takaki, and Junichi Yamagishi. "Neural source-filter waveform models for statistical parametric speech synthesis." *IEEE/ACM Transactions on Audio, Speech, and Language Processing* 28 (2019): 402-415.
>
> [2]: Choi, Hyeong-Seok, et al. "Neural analysis and synthesis: Reconstructing speech from self-supervised representations." *Advances in Neural Information Processing Systems*
>  34 (2021): 16251-16265.
>
>
> \
> \
> **Reviewer**: In section 2.2, for the linguistic feature disentanglement, what's the exact equations of breathiness perturbation?
>
> **Answer**: We are not sure what the exact equations for breathiness perturbation is as we used the high-level wrapper of the praat. We think that this can be done by extracting vocal tract filter based on a source-filter decomposition and changing the intensity of the excitation signal.

---

> > ### Author Response · Authors · 2022-11-08
> > **Response to Reviewer U62B (2/2)**
> >
> > **Review**: In section 2.3, are the global and time-varying timbre embeddings also trained in a self-supervised way or trained jointly with the waveform reconstruction? If they are jointly trained, how to guarantee the time-varying timbre embeddings don't contain other information such as pitch and linguistic?
> >
> > **Answer**: Yes, they are jointly trained. The reason why the timbre encoder does not captures other information is because the timbre encoder extracts information into a fixed number of codes regardless of the length of the target waveform. In addition, the encoding process is permutation invariant operation using cross-attention between the encoded information and trainable query vectors. Therefore, it cannot encode the sequential information.
> >
> > \
> > \
> > **Review**: In section 4.2.2, what's the problem definition here for zero-shot TTS? Does the mean the model trained on TTS task can be adapted to a new dataset or the pre-trained backbone model can be used directly for TTS task?
> >
> > **Answer**: The problem definition of zero-shot TTS is to conduct text-to-speech synthesis by transferring the timbre and prosodic style of the target utterance without additional training. The target utterances for zero-shot TTS was not used for training in both backbone model and NANSY-TTS module. So, yes, the backbone model is directly used for TTS task along with the NANSY-TTS module but it is not about adaptation.

---

### Decision · Program_Chairs · 2023-01-20

**Decision:**

Accept: poster

**Justification For Why Not Higher Score:**

The major problem of NANSY++ is that the paper needs to include ablation studies of each component. The authors proposed a set of inventions, and all of them together achieved an outstanding performance. However, we need to find out which innovation influences the results in which ways.

In addition, the performance of TTS usually highly depends on data. NANSY++ used a large in-house dataset for pre-training (the authors did use public data sets in downstream tasks, but I am talking about pre-training here). The excellent performance can even come from the enormous in-house dataset for pre-training, not related to the invention in NANSY++. Therefore, it is unclear whether the proposed approaches in NANSY++ are truly helpful or not.

From this point of view, the experiments have not yet confirmed the paper's technical contributions. So I am at the border between acceptance and rejection. "Accept (poster)" is the best I can recommend.

**Justification For Why Not Lower Score:**

I am hesitating between acceptance and rejection. NANSY++ achieves outstanding performance. However, due to the lack of an ablation study, we do not know which innovation in NANSY++ is helpful, and we need to know whether it is better than its previous version, NANSY, but I cannot find the comparison in the paper. However, using a single self-supervised model to achieve a wide range of speech synthesis tasks is novel from my viewpoint. I think it is good to let the world know that a general framework for speech synthesis based on self-supervised learning is feasible. In the end, I recommend the paper as accepted. However, I am okay with the decision if the senior meta reviewer thinks it should be rejected.

**Metareview: Summary, Strengths And Weaknesses:**

NANSY++ describes a learned analyzer and synthesizer for speech that can be trained end-to-end in a self-supervised way and demonstrates their utility in several speech synthesis tasks: voice conversion to match a target individual, voice conversion to match target characteristics, text-to-speech, and singing voice synthesis. The results of those experiments show that the proposed method does provide superior performance to the various baseline approaches. The results are impressive.

In terms of novelty, using the analyzer and synthesizer on various synthesis tasks is novel. The learning of analyzer and synthesizer is based upon an earlier work, NANSY (https://arxiv.org/abs/2110.14513). Many new components are adopted from other previous work (self-supervised pitch estimation, linguistic representation, parallel WaveGAN, and so on). The training difference between the backbone self-supervised models of NANSY and NANSY++ includes: a. NANSY++ can be end-to-end trained, b. NANSY++ directly generates waveforms, and c. time-varying timbre embeddings, etc. The authors should specify the main difference between NANSY and NANSY++ in the introduction.

A weakness of the paper is that which components in NANSY++ make it has outstanding performance is not validated. The authors' feedback does not address the issue. The authors claimed they do not have sufficient computing resources to do all the ablations. NANSY and NANSY++ have been conducted both on VC, so at least they can be compared on VC, but the authors did not do that.

The reproducibility of the paper is low. Although the authors provide details of all the architectures and hyperparameters in the appendix, the dataset for pre-training is not publicly available, and the code is not released (because of concerns about model misuse). Since NANSY++ is pre-trained from a vast high-quality in-house dataset (in downstream tasks, the authors did use public datasets), it is hard to tell the outstanding performance of NANSY++ is from the proposed approach or simply a good dataset. I suggest the authors pre-train NANSY++ on the public dataset and report the results.

I suggest the authors carefully check their baseline performances. For example, the MOS of YourTTS is close to the real speech in the original paper (https://arxiv.org/abs/2112.02418). But its MOS is far away from the real data in this paper.


**Note From Pc:**

if the above contains the word "oral" or "spotlight" please see: "oral" presentation means -> notable-top-5% and "spotlight" means -> notable-top-25%. As stated in our emails, we are disassociating presentation type from AC recommendations

**Summary Of Ac-Reviewer Meeting:**

There are four reviewers for the paper. Their scores before the meeting are as below:
U62B: update to 8 after the authors' feedback
D823: score = 8
Hzqp: score = 5
vW43: score = 5

I use a when2meet to survey the available time slots of the reviewers. We met at 10:30 p.m. on Dec 12 (GMT+8, my time zone). I can only find a time slot where D823, Hzqp and vW43 are available. So U62B did not join the meeting.

Hzqp sent me an e-mail right before the meeting that he was considering updating the score from 5 to 7 or 8. He has listened to the authors' demo in NeurIPS. He thinks the performance of NANSY++ is impressive, which changes his score on the paper. Hzqp pointed out that since he would like to change the scores, and now three reviewers want to accept the paper, we can probably cancel the meeting. However, I did not see his e-mail before the meeting.

During the discussion, in general, D823 is positive about the paper. Hzqp is at the borderline. vW43 is negative about the paper. I just let them express their opinion. I do not force them to give a conclusion.

D823 pointed out that the paper proposed a universal framework that can be applied to many applications. Even though it lacks ablations, it is good to let the world know about this paper.

D823 and Hzqp both pointed out that the paper has impressive performance, but vW43 pointed out that today many TTS models can achieve similar performance.

vW43 stated that several statements in the paper are overclaimed (he has pointed out that in his review comment). vW43 also considers the baselines problematic. The results of YourTTS has close to the actual speech performance in its original paper, but its MOS score is far from the real speech in this paper. vW43 considers the results in this paper can be misleading, so not acceptable.

Then we discuss the novelty of NANSY++ compared with NANSY. D823 considers the capability of joint training as the largest difference. vW43 considers that time timber encoding is novel. Hzqp considers the differences between NANSY and NANSY++ need to be clarified. Hzqp points out that NANSY++ is more suitable as a journal paper. First, it is very long; second, it is an extension of previous work. All reviewers agree that the paper has no fair comparison between NANSY and NANSY++.